# UGradSL: Machine Unlearning Using Gradient-based Smoothed Label

## Abstract

The objective of machine unlearning (MU) is to eliminate previously learned data from a model. However, it is challenging to strike a balance between computation cost and performance when using existing MU techniques. Taking inspiration from the influence of label smoothing on model confidence, we consider MU as decreasing confidence in the forgotten data and increasing it in the remaining. This observation suggests a simple gradient-based MU approach that uses an inverse process of label smoothing. This work introduces UGradSL, a simple, plug-and-play MU approach that uses smoothed labels. We provide theoretical analyses demonstrating why properly introducing label smoothing improves MU performance. We conducted extensive experiments on six datasets of various sizes and different modalities, demonstrating the effectiveness and robustness of our proposed method. The consistent improvement in MU performance is only at a marginal cost of additional computations. For instance, UGradSL improves over the gradient ascent MU baseline by 66% unlearning accuracy without sacrificing unlearning efficiency. This work also introduces a more practical MU paradigm, known as group-forgetting, which involves forgetting a subgroup of a superclass.

## 1 Introduction

Building a reliable ML model has become an important topic in this community. Machine unlearning (MU) is a task requiring to remove the learned data points from the model. The concept and the technology of MU enable researchers to delete sensitive or improper data in the training set to improve fairness, robustness, and privacy and get a better ML model for product usage (Chen et al., 2021; Sekhari et al., 2021). Retraining from scratch (Retrain) is a straightforward method when we want to remove the data from the model; yet it incurs prohibitive computation costs for large models due to computing resource constraints. Therefore, an efficient and effective MU method is desired.

Existing MU approaches can be generally categorized into two main categories. The first category is *exact unlearning*, which is based on retraining techniques (Bourtoule et al., 2021; Kim & Woo, 2022) and/or incorporates the principles of differential privacy (DP) (Dwork et al., 2006; Ginart et al., 2019; Guo et al., 2019; Neel et al., 2021; Ullah et al., 2021; Sekhari et al., 2021). Exact unlearning methods offer strong theoretical guarantees but are computationally intensive like Retrain. The second category is *approximate unlearning* (Koh & Liang, 2017; Golatkar et al., 2020; Warnecke et al., 2021; Graves et al., 2021; Thudi et al., 2021; Izzo et al., 2021; Becker & Liebig, 2022; Jia et al., 2023), which focuses on practical effectiveness and computational efficiency rather than providing provable guarantees. Approximate unlearning methods aim to achieve a balance between unlearning efficacy and computational complexity, making them more suitable for real-world applications.

We desire an approach that enjoys both high performance and fast speed. Since MU can be viewed as the inverse process of ML, we are motivated to think it would be a natural and efficient way to develop an unlearning process that imitates the reverse of gradient descent. Indeed, gradient ascent (GA) Thudi et al. (2021) is one of the MU methods but unfortunately, it does not fully achieve the potential of this idea. One of the primary reasons is that once the model completes training, the gradient of well-memorized data that was learned during the process is diminishing (close to 0 loss) and therefore the effect of GA is rather limited.

Our approach is inspired by the celebrated idea of label smoothing Szegedy et al. (2016). In the forward problem (gradient descent), the smoothed label proves to be able to improve the model's

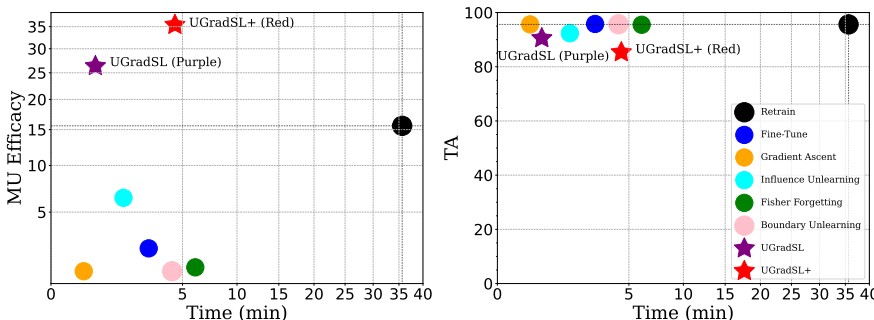

Figure 1: Summary of the proposed method and baselines (SVHN, random forgetting across all classes) in terms of the performance in forgetting dataset $D_f$ (membership inference attack), the testing dataset $D_r$ (testing accuracy) and speed. The upper left corner indicates better performance in both speed and performance. The black circle represents retrain while the circles in other colors represent the other baselines. The stars represents our methods, UGradSL (purple, ★) and UGradSL+ (red, ★). Our methods are **faster in speed and better in MU performance with little drop in the original accuracy**.

generalization power. In our setting, we show that incorporating the smoothed label in the unlearning process encourages the model to output answers with high randomness. Specifically, we show that Gradient Ascent with a "negative" label smoothing process can quickly reduce the model's confidence in the forgetting dataset, which is exactly the goal of MU. We name our approach *UGradSL*, unlearning using gradient-based smoothed labels.

One of the main highlights of our approach is that it is a plug-and-play method that can improve the gradient-based MU performance consistently and does not hurt the performance of the remaining dataset and the testing dataset in a gradient-mixed way. The framework of our method is given in Section 4.3. At the same time, we provide a theoretical analysis of the benefits of our approach for the MU task. The core contributions of this paper summarize as follows:

- We propose a lightweight tool to improve MU by joining the label smoothing and gradient ascent.
- We theoretically analyze the role of gradient ascent in MU and how negative label smoothing is able to boost the MU performance.
- Extensive experiments in six datasets in different modalities and several unlearning diagrams show the robustness and generalization of our method.
- We propose a more realistic unlearning paradigm called *group forgetting*, which can be seen as a special case of random forgetting. We conduct experiments in both benchmark and real datasets.

## 2 RELATED WORK

**Machine Unlearning**   MU was developed to address information leakage concerns related to private data after the completion of model training (Cao & Yang, 2015; Bourtoule et al., 2021; Nguyen et al., 2022), gained prominence with the advent of privacy-focused legislation (Hoofnagle et al., 2019; Pardau, 2018). One direct unlearning method involves retraining the model from scratch after removing the forgetting data from the original training set. It is computationally inefficient, prompting researchers to focus on developing approximate but much faster unlearning techniques (Becker & Liebig, 2022; Golatkar et al., 2020; Warnecke et al., 2021; Graves et al., 2021; Thudi et al., 2021; Izzo et al., 2021; Jia et al., 2023). Beyond unlearning methods, other research efforts aim to create probabilistic unlearning concepts (Ginart et al., 2019; Guo et al., 2019; Neel et al., 2021; Ullah et al., 2021; Sekhari et al., 2021) and facilitate unlearning with provable error guarantees, particularly in the context of differential privacy (DP) (Dwork et al., 2006; Ji et al., 2014; Hall et al., 2012). However, it typically necessitates stringent model and algorithmic assumptions, potentially compromising effectiveness against practical adversaries, such as membership inference attacks (Graves et al., 2021; Thudi et al., 2021). Additionally, the interest in MU has expanded to encompass

various learning tasks and paradigms (Wang et al., 2022b; Liu et al., 2022b; Chen et al., 2022; Chien et al., 2022; Marchant et al., 2022; Di et al., 2022). These diverse applications demonstrate the growing importance of MU techniques in safeguarding privacy.

**Label Smoothing**  Label smoothing (LS) or positive label smoothing (PLS) (Szegedy et al., 2016) is a commonly used regularization method to improve the model performance. Standard training with one-hot labels will lead to overfitting easily. Empirical studies have shown the effectiveness of LS in noisy label (Szegedy et al., 2016; Pereyra et al., 2017; Vaswani et al., 2017; Chorowski & Jaitly, 2016). In addition, LS shows its capability to reduce overfitting, improve generalization, etc. LS can also improve the model calibration (Müller et al., 2019). However, most of the work about LS is PLS. (Wei et al., 2021) first proposes the concept of negative label smoothing and shows there is a wider feasible domain for the smoothing rate when the rate is negative, expanding the usage of LS.

**Influence Function**  Influence function is a classic statistical method to track the impact of one training sample. (Koh & Liang, 2017) uses a second-order optimization approximation to evaluate the impact of the training sample. In addition to the single training sample, it can also be used to identify the importance of the training groups (Basu et al., 2020; Koh et al., 2019). Influence function is widely used in many machine-learning tasks. such as data bias solution (Brunet et al., 2019; Kong et al., 2021), fairness (Sattigeri et al., 2022; Wang et al., 2022a), security (Liu et al., 2022a), transfer learning (Jain et al., 2022), out-of-distribution generalization (Ye et al., 2021), etc. The approach also plays an important role as the algorithm backbone in the machine unlearning tasks (Jia et al., 2023; Warnecke et al., 2021; Izzo et al., 2021).

## 3  PROBLEM FORMULATION

**Machine Unlearning**  Consider a $K$-class classification problem on the training data distribution $\mathcal{D}_{tr} = (\mathcal{X} \times \mathcal{Y})$, where $\mathcal{X}$ and $\mathcal{Y}$ are feature and label space, respectively. Due to some privacy regulations, there exists a forgetting data distribution $\mathcal{D}_f$ that the model needs to unlearn. The MU task is to unlearn the knowledge that the model learned from the forgetting data distribution $\mathcal{D}_f$. Denote by $\mathcal{D}_r := \mathcal{D}_{tr} \backslash \mathcal{D}_f$ the retain data distribution and $\Theta_r$ the distribution of models learned on the retain distribution $\mathcal{D}_r$. Denote by $\mathcal{M}$ a learning mechanism that unlearns $\mathcal{D}_f$ from the model that learned $\mathcal{D}_{tr}$. Following (Bourtoule et al., 2021), we define the optimal solution, i.e., exact MU, as follows.

**Definition 1 (Exact MU)** *The learning mechanism $\mathcal{M}$ achieves exact machine unlearning if $\Theta_{\mathcal{M}} = \Theta_r$, where $\Theta_{\mathcal{M}}$ is the distribution of models learned using mechanism $\mathcal{M}$.*

Due to the iterative training and the sensitivity to the hyper-parameters, the current deep learning methods cannot be defined as *the exact MU*. The MU performance of retrain using iterative training is sensitive to different hyper-parameters and the distribution of the prediction on $D_f$ is not random enough as given in Section 5, showing that retrain is effectively an *approximation* of exact MU. Ideally and intuitively, the exact MU should behave as the model does not access $\mathcal{D}_f$ at all so that the prediction on $\mathcal{D}_f$ should be as random as possible.

**Label Smoothing**  In a $K$-class classification task, let $\boldsymbol{y}_i$ denote the one-hot encoded vector form of $y_i \in \mathcal{Y}$. Similar to Wei et al. (2021), we unify positive label smoothing (LS) and negative label smoothing (NLS) into generalized label smoothing (GLS). The random variable of smoothed label $\boldsymbol{y}_i^{GLS,\alpha}$ with smooth rate $\alpha \in (-\infty, 1]$ is $\boldsymbol{y}_i^{\text{GLS},\alpha} = (1 - \alpha) \cdot \boldsymbol{y}_i + \frac{\alpha}{K} \cdot \mathbf{1} = [\frac{\alpha}{K}, \cdots, \frac{\alpha}{K}, (1 + \frac{1-K}{K}\alpha), \frac{\alpha}{K}, \cdots, \frac{\alpha}{K}]$, where $(1 + \frac{1-K}{K}\alpha)$ is the $y_i$th element in the encoded label vector. When $\alpha < 0$, GLS becomes NLS.

**Basics of Influence Function**  Given a dataset $D = \{z_i : (x_i, y_i)\}_{i=1}^n$ and a function $h$ parameterized by $\boldsymbol{\theta}$ which maps from the input feature space $\mathcal{X}$ to the output space $\mathcal{Y}$. The standard empirical risk minimization writes as:

$$\boldsymbol{\theta}^* = \arg\min_{\boldsymbol{\theta}} \frac{1}{n} \sum_{z \in D} \ell(h_{\boldsymbol{\theta}}, z). \tag{1}$$

To find the impact of a training point $\hat{z}$, we up-weight its weight by an infinitesimal amount $\epsilon$. The new model parameter $\boldsymbol{\theta}_{\{\hat{z}\}}^{\epsilon}$ can be obtained from $\boldsymbol{\theta}_{\{z\}}^{\epsilon} = \arg\min_{\boldsymbol{\theta}} \frac{1}{n} \sum_{z \in D} \ell(h_{\boldsymbol{\theta}}, z) + \epsilon \cdot \ell(h_{\boldsymbol{\theta}}, \hat{z})$.

When $\epsilon = -\frac{1}{n}$, it is indicating removing $\hat{z}$. According to (Koh & Liang, 2017), $\boldsymbol{\theta}^{\epsilon}_{\{\hat{z}\}}$ can be approximated by using the first-order Taylor series expansion as

$$\boldsymbol{\theta}^{\epsilon}_{\{\hat{z}\}} \approx \boldsymbol{\theta}^* - \epsilon \cdot H_{\boldsymbol{\theta}^*}^{-1} \cdot \nabla_{\boldsymbol{\theta}} \ell\left(h_{\boldsymbol{\theta}^*}, \hat{z}\right), \tag{2}$$

where $H_{\boldsymbol{\theta}^*}$ is the Hessian with respect to (w.r.t.) $\boldsymbol{\theta}^*$. The change of $\boldsymbol{\theta}$ due to changing the weight can be given using the influence function $\mathcal{I}(\hat{z})$ as

$$\Delta\boldsymbol{\theta} = \boldsymbol{\theta}^{\epsilon}_{\{\hat{z}\}} - \boldsymbol{\theta}^* = \mathcal{I}(\hat{z}) = \left.\frac{d\boldsymbol{\theta}^{\epsilon}_{\{\hat{z}\}}}{d\epsilon}\right|_{\epsilon=0} = -H_{\boldsymbol{\theta}^*}^{-1} \cdot \nabla_{\boldsymbol{\theta}} \ell\left(h_{\boldsymbol{\theta}^*}, \hat{z}\right). \tag{3}$$

# 4 NEGATIVE LABEL SMOOTHING ENABLES FAST AND EFFECTIVE UNLEARNING

This section sets up the analysis and shows that properly performing negative label smoothing enables fast and effective unlearning. The key ingredients of our approach are gradient ascent (GA) and label smoothing (LS). We start with understanding how GA helps with unlearning and then move on to show the power of LS. At the end of the section, we formally present our algorithm.

## 4.1 GRADIENT ASCENT CAN HELP GRADIENT-BASED MACHINE UNLEARNING

We discuss three sets of model parameters in the MU problem: 1) $\boldsymbol{\theta}^*_{tr}$, the optimal parameters trained from $D_{tr} \sim \mathcal{D}_{tr}$, 2) $\boldsymbol{\theta}^*_r$, the optimal parameters trained from $D_r \sim \mathcal{D}_r$, and 3) $\boldsymbol{\theta}^*_f$, the optimal parameters unlearned using gradient ascent (GA). Note $\boldsymbol{\theta}^*_r$ can be viewed as the *exact* MU model. The definitions of $\boldsymbol{\theta}^*_{tr}$ and $\boldsymbol{\theta}^*_r$ are similar to Equation 1 and by using the influence function, $\boldsymbol{\theta}^*_f$ is

$$\boldsymbol{\theta}^*_f = \arg\min_{\boldsymbol{\theta}} R_f(\boldsymbol{\theta}) = \arg\min_{\boldsymbol{\theta}}\{R_{tr}(\boldsymbol{\theta}) + \epsilon \sum_{z^f \in D_f} \ell(h_{\boldsymbol{\theta}}, z^f)\} \tag{4}$$

where $R_{tr}(\boldsymbol{\theta}) = \sum_{z^{tr} \in D_{tr}} \ell(h_{\boldsymbol{\theta}}, z^{tr})$ is the empirical risk on $D_{tr}$. $\epsilon$ is the weight of $D_f$ compared with $D_{tr}$. The optimal parameter can be found when the gradient is 0:

$$\nabla_{\boldsymbol{\theta}} R_f(\boldsymbol{\theta}^*_f) = \nabla_{\boldsymbol{\theta}} R_{tr}(\boldsymbol{\theta}^*_f) + \epsilon \sum_{z^f \in D_f} \nabla_{\boldsymbol{\theta}} \ell(h_{\boldsymbol{\theta}^*_f}, z^f) = 0. \tag{5}$$

Expanding Equation 5 at $\boldsymbol{\theta} = \boldsymbol{\theta}^*_{tr}$ using the Taylor series, we have

$$\boldsymbol{\theta}^*_f - \boldsymbol{\theta}^*_{tr} \approx -\left[\sum_{z^{tr} \in D_{tr}} \nabla^2_{\boldsymbol{\theta}} \ell(h_{\boldsymbol{\theta}^*_{tr}}, z^{tr}) + \epsilon \sum_{z^f \in D_f} \nabla^2_{\boldsymbol{\theta}} \ell(h_{\boldsymbol{\theta}^*_{tr}}, z^f)\right]^{-1} \left(\epsilon \sum_{z^f \in D_f} \nabla_{\boldsymbol{\theta}} \ell(h_{\boldsymbol{\theta}^*_{tr}}, z^f)\right). \tag{6}$$

Similarly, we can expand $R_{tr}(\boldsymbol{\theta}^*_{tr})$ at $\boldsymbol{\theta} = \boldsymbol{\theta}^*_r$ and derive $\boldsymbol{\theta}^*_r - \boldsymbol{\theta}^*_{tr}$ as

$$\boldsymbol{\theta}^*_r - \boldsymbol{\theta}^*_{tr} \approx \left(\sum_{z^{tr} \in D_{tr}} \nabla^2_{\boldsymbol{\theta}} \ell(h_{\boldsymbol{\theta}^*_r}, z^{tr})\right)^{-1} \left(\sum_{z^{tr} \in D_{tr}} \nabla_{\boldsymbol{\theta}} \ell(h_{\boldsymbol{\theta}^*_r}, z^{tr})\right) \tag{7}$$

We ignore the average operation in the original definition of the influence function for computation convenience because the size of $D_{tr}$ or $D_f$ are fixed. For GA, let $\epsilon = -1$ in Equation 6 and we have

$$\boldsymbol{\theta}^*_r - \boldsymbol{\theta}^*_f \approx \boldsymbol{\theta}^*_r - \boldsymbol{\theta}^*_{tr} - (\boldsymbol{\theta}^*_f - \boldsymbol{\theta}^*_{tr}) = \vec{a} - \vec{b}, \qquad \text{where,} \tag{8}$$

$$\vec{a} := (\sum_{z^{tr} \in D_{tr}} \nabla^2_{\boldsymbol{\theta}} \ell(h_{\boldsymbol{\theta}^*_r}, z^{tr}))^{-1} \sum_{z^{tr} \in D_{tr}} \nabla_{\boldsymbol{\theta}} \ell(h_{\boldsymbol{\theta}^*_r}, z^{tr}), \ \ \vec{b} := (\sum_{z^r \in D_r} \nabla^2_{\boldsymbol{\theta}} \ell(h_{\boldsymbol{\theta}^*_{tr}}, z^r))^{-1} \sum_{z^f \in D_f} \nabla_{\boldsymbol{\theta}} \ell(h_{\boldsymbol{\theta}^*_{tr}}, z^f)$$

are the *true* learning phase and the *backtracked* unlearning phase, respectively. In the ideal case where the inner products of first- (gradient) and second-order (Hessian) variables of $\vec{a}$ and $\vec{b}$ are the same, GA achieves exact machine unlearning since $\boldsymbol{\theta}^*_r = \boldsymbol{\theta}^*_f$. Vector $(-\vec{a})$ is the Newton direction of learning $D_f$ (forget data) starting from $\boldsymbol{\theta}^*_r$. Vector $\vec{b}$ is the Newton direction of unlearning $D_f$ starting from $\boldsymbol{\theta}^*_{tr}$. However, in practice, due to the inconsistency between $\boldsymbol{\theta}^*_{tr}$ and $\boldsymbol{\theta}^*_r$, $\|\boldsymbol{\theta}^*_r - \boldsymbol{\theta}^*_f\|$ cannot be naively treated as 0, i.e., GA cannot always help MU. We summarize it in Theorem 1.

**Theorem 1** *Given the approximation in Equation 8, GA achieve exact MU if and only if*

$$\sum_{z^f \in D_f} \nabla_{\boldsymbol{\theta}} \ell(h_{\boldsymbol{\theta}_r^*}, z^f) = -\boldsymbol{H}(\boldsymbol{\theta}_r^*, \boldsymbol{\theta}_{tr}^*) \cdot \sum_{z^f \in D_f} \nabla_{\boldsymbol{\theta}} \ell(h_{\boldsymbol{\theta}_{tr}^*}, z^f),$$

*where $\boldsymbol{H}(\boldsymbol{\theta}_r^*, \boldsymbol{\theta}_{tr}^*) = \left(\sum_{z^{tr} \in D_{tr}} \nabla_{\boldsymbol{\theta}}^2 \ell(h_{\boldsymbol{\theta}_r^*}, z^{tr})\right) \left(\sum_{z^r \in D_r} \nabla_{\boldsymbol{\theta}}^2 \ell(h_{\boldsymbol{\theta}_{tr}^*}, z^r)\right)^{-1}$. Otherwise, there exist $\boldsymbol{\theta}_r^*, \boldsymbol{\theta}_{tr}^*$ such that GA can not help MU, i.e., $\|\boldsymbol{\theta}_r^* - \boldsymbol{\theta}_f^*\| > \|\boldsymbol{\theta}_r^* - \boldsymbol{\theta}_{tr}^*\|$.*

## 4.2 NEGATIVE LABEL SMOOTHING IMPROVES MU

In practice, we cannot guarantee that GA always helps MU as shown in Theorem 1. To alleviate the possible undesired effect of GA, we propose to use label smoothing as a plug-in module. Consider the cross-entropy loss as an example. For GLS, the loss is calculated as

$$\ell(h_{\boldsymbol{\theta}}, z^{\text{GLS},\alpha}) = \left(1 + \frac{1-K}{K}\alpha\right) \cdot \ell(h_{\boldsymbol{\theta}}, (x,y)) + \frac{\alpha}{K} \sum_{y' \in \mathcal{Y} \setminus y} \ell(h_{\boldsymbol{\theta}}, (x,y')), \tag{9}$$

where we use notations $\ell(h_{\boldsymbol{\theta}}, (x,y)) := \ell(h_{\boldsymbol{\theta}}, z)$ to specify the loss of an example $z = (x,y)$ in the dataset and $\ell(h_{\boldsymbol{\theta}}, (x,y))$ to denote the loss of an example when its label is replaced with $y'$.

**Theorem 2** *Given the approximation in Equation 8 and $\langle \vec{a} - \vec{b}, \vec{c} - \vec{b} \rangle \leq 0$, there exists an $\alpha < 0$ such that NLS helps GA, i.e.,*

$$\text{if } \langle \vec{a} - \vec{b}, \vec{c} - \vec{b} \rangle \leq 0: \ \exists \alpha < 0, \ \|\boldsymbol{\theta}_r^* - \boldsymbol{\theta}_{f,GLS}^*\| < \|\boldsymbol{\theta}_r^* - \boldsymbol{\theta}_f^*\|, \qquad \text{where,}$$

- $\vec{c} := \frac{1}{K-1} \left(\sum_{z^r \in D_r} \nabla_{\boldsymbol{\theta}}^2 \ell(h_{\boldsymbol{\theta}_{tr}^*}, z^r)\right)^{-1} \sum_{z^f \in D_f} \nabla_{\boldsymbol{\theta}} \sum_{y' \in \mathcal{Y} \setminus y^f} \ell(h_{\boldsymbol{\theta}_{tr}^*}, (x^f, y'))$, *capturing the gradient influence of the smoothed non-target label on the weight.*

- $(-\vec{a}) + \vec{b}$ *is the resultant of Newton direction of learning and unlearning.*

- $(-\vec{c}) + \vec{b}$ *is resultant of Newton direction of learning non-target labels and unlearning the target label.*

**How the smoothed term works** Intuitively, Term $\sum_{y' \in \mathcal{Y} \setminus y} \ell(h_{\boldsymbol{\theta}}, (x,y'))$ leads to a state where the model makes wrong predictions on data in the forgetting dataset with equally low confidence (Wei et al., 2021; Lukasik et al., 2020). If the exact MU state does not overfit any points in the forgetting dataset and takes random guesses, then Term $\sum_{y' \in \mathcal{Y} \setminus y} \ell(h_{\boldsymbol{\theta}}, (x,y'))$ directs the model to be closer to the exact MU. On the other hand, Term $(- \sum_{y' \in \mathcal{Y} \setminus y} \ell(h_{\boldsymbol{\theta}}, (x,y')))$ leads to a state where the model randomly picks one wrong prediction for each data point in the forgetting dataset with high confidence (Cheng et al., 2021; Liu & Guo, 2020). If the exact MU state overfits the forgetting dataset with wrong labels, then Term $(-\vec{c})$ directs the model to be closer to the exact MU. Denote by $\boldsymbol{\theta}_{f,GLS}^*$ the optimal parameters unlearned using GA and NLS, and $\langle \cdot, \cdot \rangle$ the inner product of two vectors. We show the effect of NLS in Theorem 2.

Now we briefly discuss when the condition $\langle \vec{a} - \vec{b}, \vec{c} - \vec{b} \rangle \leq 0$ holds. $\vec{c} - \vec{b}$ roughly speaking captures the effects of the smoothing term in the unlearning process. If we assume that the exact MU model is able to fully unlearn the example and outputs random predictions, vector $\vec{c}$ contributes a direction that pushes the model closer to the exact MU state, therefore reducing $\|\vec{a} - \vec{b}\|$ - this will correspond to a negative inner product between the two terms.

## 4.3 UGRADSL: A PLUG-AND-PLAY AND GRADIENT-MIXED MU METHOD

Compared with Retrain, Fine-Tune (FT) and GA are much more efficient as illustrated in the Experiment part in Section 5 with comparable or better MU performance. FT and GA focus on different perspectives of MU. FT is to transfer the knowledge of the model from $D_{tr}$ to $D_r$ using gradient descent (GD) while GA is to remove the knowledge of $D_f$ from the model. As a plug-and-play algorithm, our method is suitable for the gradient-based methods including FT and GA. UGradSL is based on GA while UGradSL+ is on FT. Compared with UGradSL, UGradSL+ will lead to a more comprehensive result but with a larger computation cost.

We present UGradSL and UGradSL+ in Algorithm 1. For UGradSL+, we first sample a batch $B_r = \{z_i^r : (x_i^r, y_i^r)\}_{i=1}^{n_{B_r}}$ from $D_r$ (Line 3-4). Additionally, we sample a batch $B_f = \{z_i^f : (x_i^f, y_i^f)\}_{i=1}^{n_{B_f}}$

from $D_f$ where $n_{B_r} = n_{B_f}$ (Line 5). It is very likely that $D_f$ will be iterated several times when $D_r$ is fully iterated once because $n_r > n_f$ in general. Then we apply NLS on $B_f$ leading to $B_f^{\text{NLS},\alpha} = \{z_i^{f,\text{NLS},\alpha} : (x_i^f, y_i^{f,\text{NLS},\alpha})\}$. We calculate the loss using a gradient-mixed method as:

$$L(h_{\boldsymbol{\theta}}, B_f^{\text{NLS},\alpha}, B_r, p) = p \cdot \sum_{z^r \in B_r} \ell(h_{\boldsymbol{\theta}}, z^r) - (1-p) \cdot \sum_{z^{f,\text{NLS},\alpha} \in B_f^{\text{NLS},\alpha}} \ell(h_{\boldsymbol{\theta}}, z^{f,\text{NLS},\alpha}), \quad (10)$$

where $p \in [0,1]$ is used to balance GD and GA and the minus sign stands for the GA. $h_{\boldsymbol{\theta}}$ is updated according to $L$ (Line 6). UGradSL is similar to UGradSL+ and the dataset used is given in bracket in Algorithm 1. The difference between UGradSL and UGradSL+ is the convergence standard. UGradSL is based on the convergence of $D_f$ while UGradSL+ is based on $D_r$. It should be noted that the Hessian matrix in Theorem 1 is only used in the theoretical proof. In the practical calculation, **there is no need to calculate the Hessian matrix**. Thus, our method does not incur substantially more computation but improves the MU performance on a large scale. We present empirical evidence in Section 5.

---

**Algorithm 1** UGradSL+: A plug-and-play, efficient, gradient-based MU method using NLS. UGradSL can be specified by imposing the dataset replacement in the bracket.

---

**Require:** A almost-converged model $h_{\hat{\boldsymbol{\theta}}_{tr}}$ trained with $D_{tr}$. The retained dataset $D_r$. The forgetting dataset $D_f$. The smoothing rate $\alpha$. Unlearning epochs $E$. GA ratio $p$.
**Ensure:** The unlearned model $h_{\boldsymbol{\theta}_f}$.
1: Set the current epoch index as $t_c \leftarrow 1$
2: **while** $t_c < E$ **do**
3:     **while** $D_r(D_f)$ is not fully iterated **do**
4:         Sample a batch $B_r$ in $D_r$
5:         Sample a batch $B_f$ from $D_f$ where the size of $B_f$ is the same as that of $B_r$
6:         Update the model using $B_r$, $B_f$, $p$ and $\alpha$ according to Equation 10
7:     **end while**
8:     $t_c \leftarrow t_c + 1$
9: **end while**

---

## 5 EXPERIMENTS AND RESULTS

### 5.1 EXPERIMENT SETUP

**Dataset and Model Selection** We validate our method using various datasets in different scales and modality, including CIFAR-10 (Krizhevsky et al., 2009), CIFAR-100 (Krizhevsky et al., 2009), SVHN (Netzer et al., 2011), CelebA (Liu et al., 2015), ImageNet (Deng et al., 2009) and 20 Newsgroup (Caldas et al., 2018) datasets. For the vision and language dataset, we use ResNet-18 (He et al., 2016) and Bert (Devlin et al., 2018) as the backbone model, respectively. Due to the page limit, the details of the training parameter and the additional results of different models including VGG-16 and vision transformer (ViT) are given in the Appendix.

**Baseline Methods** We compare UGradSL and UGradSL+ with a series of baseline methods, including retrain, fine-tuning (FT) (Warnecke et al., 2021; Golatkar et al., 2020), gradient ascent (GA) (Graves et al., 2021; Thudi et al., 2021), fisher forgetting (FF) (Becker & Liebig, 2022; Golatkar et al., 2020), unlearning based on the influence function (IU) (Izzo et al., 2021; Koh & Liang, 2017) and boundary unlearning (BU) (Chen et al., 2023). Besides, there is also an unlearning paradigm called instance unlearning (Cha et al., 2023), which is not the scope in this paper.

**Evaluation Metrics** There are three groups of evaluation metrics. The *first* group of evaluation metrics follows (Jia et al., 2023), where we jointly consider unlearning accuracy (UA), membership inference attack (MIA), remaining accuracy (RA), testing accuracy (TA), and run-time efficiency (RTE). UA is the ratio of incorrect prediction on $D_f$, showing the MU performance. The higher UA is, the better the MU performance is. MIA is to evaluate the ratio of data points in $D_f$ belonging to the training set of the unlearned model $\boldsymbol{\theta}_u$. The higher MIA is, the less information of $D_f$ is included in $\boldsymbol{\theta}_u$. RA is the accuracy on $D_r$. TA is the accuracy used to evaluate the performance on the whole

testing set, except for the class-wise forgetting because the task is to forget the specific class. Note a *tradeoff* between RA/TA and UA/MIA exists, i.e., the higher UA/MIA usually implies lower RA/TA in practice. In our experiments, we expect RA and TA will not decrease too much (e.g., 5%) while MA and MIA can improve consistently and more significantly.

The *second* evaluation metric relies on information entropy (Shannon, 2001). As mentioned in Section 3, the prediction on $D_f$ from a good MU model should be as random as possible, implying little enough information about $D_f$. Therefore, we introduce $H$ by first calculating the sample-wise entropy of model-predicted probabilities on each data point in $D_f$, then taking an average, which is the higher the better. The *other* evaluation metric is RTE, which measures the time of MU. We expect a smaller RTE if the other two groups of metrics are similar. All results are averaged by running 3 experiments with different random seeds. The selection of $D_f$ is the same for the same seed in the corresponding dataset.

**Unlearning Diagram**  We mainly consider three unlearning diagrams, including *class-wise forgetting*, *random forgetting across all classes*, and *group forgetting*. Class-wise forgetting is to unlearn the whole specific class where we remove one class in $D_r$ and the corresponding class in the testing dataset completely. Random forgetting across all classes is to unlearn data points belonging to all classes. In addition to random forgetting across all classes, random forgetting within one class is discussed in (Jia et al., 2023; Golatkar et al., 2020; Graves et al., 2021). The results of random forgetting within one class are given in Appendix due to the page limit. As a special case of random forgetting, we propose a new setting called *group forgetting*, meaning that the model is trained to unlearn the group or sub-class of the corresponding super-classes. There are many practical cases of this setting. For example, due to the copyright or privacy issue, we need to unlearn the specific identities from a model classifying the facial attributes. We believe this setting has a more practical meaning which will broaden the usage of MU.

## 5.2 CLASS-WISE FORGETTING

We select the class randomly and run class-wise forgetting on four datasets. We report the results of CIFAR-10, ImageNet and 20 Newsgroup in Table 1 due to the page limit. The results of CIFAR-100 and SVHN are given in Appendix. As we can see, UGradSL and UGradSL+ can boost the performance of GA and FT, respectively without an increment in RTE or drop in TA and RA, leading to comprehensive satisfaction in the main metrics, even in the randomness on $D_f$, showing the robustness and flexibility of our methods in MU.

## 5.3 RANDOM FORGETTING ACROSS ALL CLASSES

We select data randomly from every class as $D_f$, making sure all the classes are selected and the size of $D_f$ is 10% of the $D_{tr}$. We report the results of CIFAR-100 and 20 Newsgroup in Table 2. Compared with class-wise forgetting, it is harder to improve the MU performance without a significant drop in the remaining accuracy in random forgetting across all dataset. Our methods are much better in UA and MIA than the baseline methods, even retrain. Compared with FT and GA, UGradSL+ can improve the UA or MIA by more than 50% with a drop in RA or TA by 15% at most. We also run the experiments of CIFAR-10, SVHN and ImageNet which are given in Appendix.

## 5.4 GROUP FORGETTING

Although group forgetting can be seen as part of random forgetting within one class or across all classes, we want to highlight its use case here due to its practical impacts on *e.g.*, facial attributes classification. The identities can be regarded as the subgroup in the attributes.

### 5.4.1 GROUP FORGETTING WITHIN ONE CLASS ON CIFAR-100

CIFAR-10 and CIFAR-100 share the same image dataset while CIFAR-100 is labeled with 100 sub-classes and 20 super-classes (Krizhevsky et al., 2009). We train a model to classify 20 super-classes using CIFAR-100 training set. The setting of the *group forgetting within one super-class* is to remove one sub-class from one super-class in CIFAR-100 datasets. For example, there are five fine-grained fishes in the *Fish* super-class and we want to remove one fine-grained fish from the model. Different

Table 1: Results of class-wise forgetting in CIFAR-10, 20 Newsgroup and ImageNet.

| CIFAR-10 | UA (↑) | MIA (↑) | RA (↑) | TA (↑) | $H$ (↑) | RTE (↓, min) |
|---|---|---|---|---|---|---|
| Retrain | 100.00±0.00 | 100.00±0.00 | 98.19±3.14 | 94.50±0.34 | 0.27 | 24.62 |
| FT | 22.71±5.31 | 79.21±8.60 | **99.82±0.09** | **94.13±0.14** | 0.79 | 2.02 |
| GA | 25.19±11.38 | 73.48±9.68 | 96.84±0.58 | 73.10±1.62 | 0.27 | **0.08** |
| IU | 83.92±1.16 | 92.59±1.41 | 98.77±0.12 | 92.64±0.23 | 0.30 | 1.18 |
| FF | 5.28±2.22 | 12.90±5.51 | 92.83±2.68 | 92.64±2.68 | 1.14 | 6.75 |
| BU | 78.33±3.47 | 92.63±2.19 | 97.28±0.99 | 90.93±0.81 | 0.78 | 1.42 |
| UGradSL | **91.33±7.52** | 91.87±7.06 | 85.97±5.98 | 81.30±4.99 | **1.43** | 0.22 |
| UGradSL+ | 88.24±2.87 | **99.63±1.69** | 98.88±0.33 | 92.46±0.47 | 0.88 | 3.07 |

| 20 Newsgroup | UA (↑) | MIA (↑) | RA (↑) | TA (↑) | $H$ (↑) | RTE (↓, min) |
|---|---|---|---|---|---|---|
| Retrain | 100.00±0.00 | 100.00±0.00 | 98.31±2.56 | 81.95±1.69 | 0.52 | 26.246 |
| FT | 4.14±2.11 | 9.23±3.40 | 98.83±0.86 | 82.63±0.73 | 0.13 | 1.77 |
| GA | 17.12±9.48 | 62.03±5.84 | 99.99±0.01 | 85.41±0.37 | 0.17 | **0.37** |
| IU | 0.00±0.00 | 0.25±0.12 | **100.00±0.00** | **85.58±0.20** | 0.003 | 1.52 |
| FF | 5.08±1.72 | 15.34±7.78 | 96.98±1.23 | 79.08±0.88 | 0.31 | 13.35 |
| UGradSL | **100.00±0.00** | **100.00±0.00** | 96.31±4.02 | 78.54±5.10 | **0.78** | 0.39 |
| UGradSL+ | **100.00±0.00** | **100.00±0.00** | 99.76±0.23 | 84.21±0.41 | 0.01 | 2.13 |

| ImageNet | UA (↑) | MIA (↑) | RA (↑) | TA (↑) | $H$ (↑) | RTE (↓, hr) |
|---|---|---|---|---|---|---|
| Retrain | 100.00±0.00 | 100.00±0.00 | 71.62±0.12 | 69.57±0.07 | 1.20 | 26.18 |
| FT | 52.42±15.81 | 55.87±18.02 | 70.66±2.54 | **69.25±0.78** | 2.20 | 2.87 |
| GA | 81.23±0.69 | 83.52±2.08 | 66.00±0.03 | 64.72±0.02 | 2.02 | **0.01** |
| IU | 33.54±19.46 | 49.83±21.57 | 66.25±1.99 | 66.28±1.19 | 0.25 | 1.51 |
| FF | - | - | - | - | - | 145 |
| UGradSL | **100.00±0.00** | **100.00±0.00** | 76.91±1.82 | 65.94±1.35 | 3.03 | **0.01** |
| UGradSL+ | **100.00±0.00** | **100.00±0.00** | **78.16±0.07** | 66.84±0.06 | **3.49** | 4.19 |

Table 2: Results of random forgetting across all classes in CIFAR-100 and 20 Newsgroup.

| CIFAR-100 | UA (↑) | MIA (↑) | RA (↑) | TA (↑) | $H$ (↑) | RTE (↓, min) |
|---|---|---|---|---|---|---|
| Retrain | 29.47±1.59 | 53.50±1.19 | 99.98±0.01 | 70.51±1.17 | 0.76 | 25.01 |
| FT | 2.55±0.03 | 10.59±0.27 | **99.95±0.01** | 75.95±0.05 | 0.18 | 1.95 |
| GA | 2.58±0.06 | 5.95±0.17 | 97.45±0.02 | **76.09±0.01** | 0.12 | **0.29** |
| IU | 15.71±5.19 | 18.69±4.12 | 84.65±5.29 | 62.20±4.17 | 0.25 | 1.20 |
| FF | 5.55±4.94 | 11.04±6.68 | 93.52±5.60 | 70.58±6.33 | 0.32 | 42.75 |
| BU | 2.38±0.14 | 5.95±0.09 | 97.43±0.03 | 76.17±0.01 | 0.12 | 2.4 |
| UGradSL | 15.10±2.76 | 34.67±0.63 | 86.69±2.41 | 59.25±2.35 | **1.67** | 0.55 |
| UGradSL+ | **63.89±0.75** | **71.51±1.31** | 92.25±0.11 | 61.09±0.10 | 1.11 | 3.52 |

| 20 Newsgroup | UA (↑) | MIA (↑) | RA (↑) | TA (↑) | $H$ (↑) | RTE (↓, min) |
|---|---|---|---|---|---|---|
| Retrain | 7.37±0.14 | 9.33±0.98 | 100±0.01 | 85.24±0.09 | 0.03 | 75.12 |
| FT | 2.26±1.53 | 2.70±1.60 | 98.60±0.18 | 82.20±1.12 | 0.02 | 1.06 |
| GA | 0.74±0.97 | 2.65±0.98 | 99.69±0.14 | 83.63±0.12 | 0.01 | **0.77** |
| IU | 0.03±0.06 | 0.33±0.11 | **100.00±0.00** | **85.72±0.12** | 0.004 | 1.53 |
| FF | 1.57±1.07 | 5.68±2.34 | 98.17±0.74 | 80.32±1.46 | 0.14 | 16.88 |
| UGradSL | 13.00±1.17 | 44.3±1.36 | 93.46±1.01 | 73.17±0.42 | **2.54** | 1.50 |
| UGradSL+ | **35.53±1.53** | **46.42±0.49** | 97.15±0.88 | 78.95±1.98 | 0.83 | 4.73 |

from class-wise forgetting, we do not modify the testing set. We report the group forgetting within one super-class in Table 3. The details and results of the *group forgetting across all super-classes* are given in Appendix.

### 5.4.2 GROUP FORGETTING ON CELEBA

We select CelebA dataset as another real-world case and show the results in Table 4. We train a binary classification model to classify whether the person is smile or not. There are 8192 identities in the training set and we select 1% of the identities (82 identies) as $D_f$. Both smiling and non-smiling

Table 3: The experiment results of group forgetting within one class using the CIFAR-100 dataset. The model is classify 20 super-classes and $D_f$ is one of five subclasses in one super-class.

| | UA (↑) | MIA (↑) | RA (↑) | TA (↑) | $H$ (↑) | Time (↓, min) |
|---|---|---|---|---|---|---|
| Retrain | 74.22±1.39 | 88.07±2.94 | 99.93±0.44 | 82.57±1.19 | 0.75 | 27.35 |
| FT | 6.00±0.61 | 24.96±2.90 | **99.55±0.30** | **82.94±0.55** | 0.42 | 7.47 |
| GA | 87.19±0.55 | **93.85±2.28** | 91.04±2.70 | 74.64±1.78 | 0.95 | **0.11** |
| IU | 18.00±15.09 | 51.67±20.58 | 98.19±0.24 | 82.72±0.94 | 0.82 | 1.10 |
| FF | 4.00±5.21 | 13.41±12.29 | 97.14±1.07 | 81.24±1.32 | 0.45 | 3.81 |
| BU | 78.37±0.46 | 81.33±2.91 | 23.14±0.10 | 22.57±0.02 | **2.34** | 0.29 |
| UGradSL | **88.00±1.06** | 93.63±1.46 | 95.06±1.19 | 78.58±1.17 | 1.46 | 0.13 |
| UGradSL+ | 81.11±0.74 | 86.96±1.89 | 95.69±0.82 | 78.02±1.02 | 1.37 | 8.12 |

Table 4: The experiment results of group forgetting in CelebA. The model is to classify whether the person is smile or not and $D_f$ is selected according to the identities.

| | UA (↑) | MIA (↑) | RA (↑) | TA (↑) | $H$ (↑) | Time (↓, min) |
|---|---|---|---|---|---|---|
| Retrain | 6.74±0.26 | 9.77±1.49 | 94.38±0.49 | 91.78±0.33 | 0.14 | 258.69 |
| FT | 5.36±0.17 | 5.87±0.11 | **93.91±0.04** | **93.18±0.03** | 0.15 | 25.94 |
| GA | 6.00±0.16 | 5.76±0.14 | 92.86±0.13 | 92.52±0.08 | 0.17 | **1.20** |
| IU | 5.90±0.11 | 4.91±0.30 | 93.05±0.01 | 92.62±0.01 | 0.19 | 219.77 |
| FF | 5.79±0.03 | 5.08±0.07 | 93.05±0.01 | 92.60±0.02 | 0.19 | 217.02 |
| UGradSL | 11.33±4.17 | 23.08±11.53 | 87.86±3.85 | 87.68±3.81 | **0.46** | 2.17 |
| UGradSL+ | **15.63±8.01** | **26.95±25.80** | 89.17±5.86 | 88.29±5.75 | 0.23 | 51.41 |

images are in $D_f$. This experiment has significant practical meaning, since the bio-metric, such as identity and fingerprint, needs more privacy protection (Minaee et al., 2023). Compared with baseline methods, our method can forget the identity information better without forgetting too much remaining information in the dataset. This paradigm provides a practical usage of MU and our methods provide a faster and more reliable way to improve the MU performance.

## 5.5 THE DISCUSSION OF THE PERFORMANCE TRADEOFF

As shown in Table 1, while we achieve strong unlearning performance there is no apparent drop of the remaining performance (RA, TA) for the class-wise forgetting compared with other baselines. For random forgetting and group forgetting, the boost of the forgetting performance (UA, MIA) inevitably leads to drops in the remaining performance as shown in Table 2, 3, 4. This is a commonly observed trade-off in the MU literature (Jia et al., 2023; Warnecke et al., 2021; Graves et al., 2021). Yet we would like to note that at the mild cost of the drops, we observe significant improvement in unlearning performance. For example, in Table 2, the remaining performance drop in CIFAR-100 are 7.73 (TA) and 9.42 (RA), respectively compared with retrain. However, our unlearning performance boosts are 48.18 (UA) and 52.82 (MIA), respectively compared with the best baseline methods.

Note the proposed method also works well when $D_r$ is not used in the algorithm. See Appendix **??** for the ablation study. We also show more experiments to Appendix D.1

## 6 CONCLUSIONS

We have proposed UGradSL, a plug-and-play, efficient, gradient-based MU method using smoothed labels. Theoretical proofs and extensive numerical experiments have demonstrated the effectiveness of the proposed method. We have also proposed a new unlearning paradigm called group forgetting which has more practical meaning. Our work has limitations. For example, we desire an efficient way to find the exact MU state in experiments and further explore the applications of MU to promote privacy and fairness. Our method can be further validated and tested in other tasks, such as unlearning recommendation systems, etc.

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

Table 5: Notation used in this paper

| Notations | Description |
|---|---|
| $K$ | The number of class in the dataset |
| $\mathcal{D}, \mathcal{X}, \mathcal{Y}$ | The general dataset distribution, the feature space and the label space |
| $D$ | The dataset $D \in \mathcal{D}$ |
| $D_{tr}, D_r, D_f$ | The training set, remaining set and forgetting set |
| $\Theta_{\mathcal{M}}$ | The distribution of models learned using mechanism $\mathcal{M}$ |
| $\boldsymbol{\theta}$ | The model weight |
| $\boldsymbol{\theta}^*$ | The optimal model weight |
| $\boldsymbol{\theta}^*_{f,\text{LS}}$ | The optimal model weight trained with $D_f$ whose label is smoothed |
| $\|\boldsymbol{\theta}\|$ | The 2-norm of the model weight |
| $n$ | The size of dataset |
| $\epsilon$ | The up-weighted weight of datapoint $z$ in influence function |
| $\mathcal{I}(z)$ | Influence function of data point $z$ |
| $h_{\boldsymbol{\theta}}$ | A function $h$ parameterized by $\boldsymbol{\theta}$ |
| $\ell(h_{\boldsymbol{\theta}}, z_i)$ | Loss of $h_{\boldsymbol{\theta}}(x_i)$ and $y_i$ |
| $R_{tr}(\boldsymbol{\theta})$ | The empirical risk of training set when the model weight is $\boldsymbol{\theta}$ |
| $R_f(\boldsymbol{\theta})$ | The empirical risk of forgetting set when the model weight is $\boldsymbol{\theta}$ |
| $R_r(\boldsymbol{\theta})$ | The empirical risk of remaining set when the model weight is $\boldsymbol{\theta}$ |
| $H_{\boldsymbol{\theta}}$ | The Hessian matrix w.r.t. $\boldsymbol{\theta}$ |
| $\nabla_{\boldsymbol{\theta}}$ | The gradient w.r.t. $\boldsymbol{\theta}$ |
| $B$ | Data batch |
| $B^{\text{LS},\alpha}$ | The smoothed batch using $\alpha$ |
| $z_i = (x_i, y_i)$ | A data point $z_i$ whose feature is $x_i$ and label is $y_i$ |
| $\boldsymbol{y}_i$ | The one-hot encoded vector form of $y_i$ |
| $\boldsymbol{y}_i^{\text{GLS},\alpha}$ | The smoothed one-hot encoded vector form of $y_i$ where the smooth rate is $\alpha$ |
| $\alpha$ | Smooth rate in general label smoothing |

**Roadmap** The appendix is composed as follows. Section A presents all the notations and their meaning we use in this paper. Section B shows the pipeline of our methods. Section C gives the proof of our theoretical analysis. Section D shows the additional experiment results with more details that are not given in the main paper due to the page limit.

## A  NOTATION TABLE

The notations we use in the paper is summaried in the Table 5.

## B  THE FRAMEWORK OF OUR METHOD

Our framework is shown in given 2. We only apply NLS on the forgetting dataset $D_f$. In back-propagation process, we apply gradient descent on the data $z_i^r \in D_r$ and gradient ascent on the data smoothed $D_f$, which is the mix-gradient way.

## C  PROOFS

### C.1  PROOF FOR THEOREM 1

For $p(x)$, the Taylor expansion at $x = a$ is

$$p(x) = p(a) + \frac{p'(a)}{1}(x - a) + o \tag{11}$$

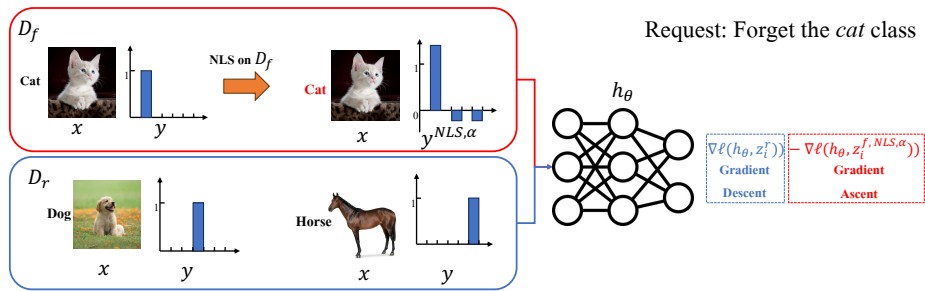

Figure 2: The framework of UGradLS. When there is a unlearning request, we can split the $D_{tr}$ into $D_f$ and $D_r$. We first apply NLS on $z_i^f = \{x, y\} \in D_f$ to get $z_i^{\mathrm{NLS}, \alpha} = \{x, y^{\mathrm{NLS}, \alpha}\}$. In back-propagation process, we apply gradient descent on the data $z_i^r \in D_r$ and gradient ascent on the data smoothed $D_f$, which is the mix-gradient way.

Here $p(\boldsymbol{\theta}) = \nabla R_{tr}(\boldsymbol{\theta}) + \epsilon \sum_{D_f} \nabla \ell(h_{\boldsymbol{\theta}}, z_i^f)$ so we have

$$p(\boldsymbol{\theta}) = \nabla R_{tr}(a) + \epsilon \sum_{z^f \in D_f} \nabla \ell(h_a, z^f) + (\nabla^2 R_{tr}(a) + \epsilon \sum_{z^f \in D_f} \nabla^2 \ell(h_a, z^f))(\boldsymbol{\theta} - a) + o \quad (12)$$

For Equation 5, we expand $f(\boldsymbol{\theta}_f)$ at $\boldsymbol{\theta} = \boldsymbol{\theta}_{tr}^*$ as

$$p(\boldsymbol{\theta}_f^*) = \nabla R_{tr}(\boldsymbol{\theta}_{tr}^*) + \epsilon \sum_{z^f \in D_f} \nabla \ell(h_{\boldsymbol{\theta}_{tr}^*}, z^f) + \left[ \nabla^2 R_{tr}(\boldsymbol{\theta}_{tr}^*) + \epsilon \sum_{z^f \in D_f} \nabla^2 \ell(h_{\boldsymbol{\theta}_{tr}^*}, z^f) \right](\boldsymbol{\theta}_f^* - \boldsymbol{\theta}_{tr}^*) + o = 0$$

$$\nabla R_{tr}(\boldsymbol{\theta}_{tr}^*) + \epsilon \sum_{z^f \in D_f} \nabla \ell(h_{\boldsymbol{\theta}_{tr}^*}, z^f) + \left[ \nabla^2 R_{tr}(\boldsymbol{\theta}_{tr}^*) + \epsilon \sum_{z^f \in D_f} \nabla^2 \ell(h_{\boldsymbol{\theta}_{tr}^*}, z^f) \right](\boldsymbol{\theta}_f^* - \boldsymbol{\theta}_{tr}^*) \approx 0$$

$$-\left[ \nabla^2 R_{tr}(\boldsymbol{\theta}_{tr}^*) + \epsilon \sum_{z^f \in D_f} \nabla^2 \ell(h_{\boldsymbol{\theta}_{tr}^*}, z^f) \right]^{-1} \left[ \nabla R_{tr}(\boldsymbol{\theta}_{tr}^*) + \epsilon \sum_{z^f \in D_f} \nabla \ell(h_{\boldsymbol{\theta}_{tr}^*}, z^f) \right] = (\boldsymbol{\theta}_f^* - \boldsymbol{\theta}_{tr}^*)$$

$$(13)$$

We have $\nabla R_{tr}(\boldsymbol{\theta}_{tr}^*) = 0$ and

$$\boldsymbol{\theta}_f^* - \boldsymbol{\theta}_{tr}^* = -\left[ \nabla^2 R_{tr}(\boldsymbol{\theta}_{tr}^*) + \epsilon \sum_{z^f \in D_f} \nabla^2 \ell(h_{\boldsymbol{\theta}_{tr}^*}, z^f) \right]^{-1} (\epsilon \sum_{z^f \in D_f} \nabla \ell(h_{\boldsymbol{\theta}_{tr}^*}, z^f)$$

$$= -\left[ \sum_{z^{tr} \in D_{tr}} \nabla^2 \ell(h_{\boldsymbol{\theta}_{tr}^*}, z^{tr}) + \epsilon \sum_{z^f \in D_f} \nabla^2 \ell(h_{\boldsymbol{\theta}_{tr}^*}, z^f) \right]^{-1} (\epsilon \sum_{z^f \in D_f} \nabla \ell(h_{\boldsymbol{\theta}_{tr}^*}, z^f) \quad (14)$$

We expand $q(\boldsymbol{\theta}_{tr}^*)$ at $\boldsymbol{\theta} = \boldsymbol{\theta}_r^*$ as

$$q(\boldsymbol{\theta}_{tr}^*) = \sum_{z^{tr} \in D_{tr}} \nabla \ell(h_{\boldsymbol{\theta}_r^*}, z^{tr}) + (\sum_{z^{tr} \in D_{tr}} \nabla^2 \ell(h_{\boldsymbol{\theta}_r^*}, z^{tr})(\boldsymbol{\theta}_{tr}^* - \boldsymbol{\theta}_r^*) \approx 0$$

$$\boldsymbol{\theta}_r^* - \boldsymbol{\theta}_{tr}^* = \left( \sum_{z^{tr} \in D_{tr}} \nabla^2 \ell(h_{\boldsymbol{\theta}_r^*}, z^{tr}) \right)^{-1} \sum_{z^{tr} \in D_{tr}} \nabla \ell(h_{\boldsymbol{\theta}_r^*}, z^{tr}) \quad (15)$$

Because of gradient ascent, $\epsilon = -1$ and we have

$$
\begin{aligned}
\boldsymbol{\theta}_r^* - \boldsymbol{\theta}_f^* &= \boldsymbol{\theta}_r^* - \boldsymbol{\theta}_{tr}^* - (\boldsymbol{\theta}_{tr}^* - \boldsymbol{\theta}_f^*) = \left( \sum_{z^{tr} \in D_{tr}} \nabla^2 \ell(h_{\boldsymbol{\theta}_r^*}, z^{tr}) \right)^{-1} \sum_{z^{tr} \in D_{tr}} \nabla \ell(h_{\boldsymbol{\theta}_r^*}, z^{tr}) \\
&\quad - \left[ \sum_{z^{tr} \in D_{tr}} \nabla^2 \ell(h_{\boldsymbol{\theta}_{tr}^*}, z^{tr}) - \sum_{z^f \in D_f} \nabla^2 \ell(h_{\boldsymbol{\theta}_{tr}^*}, z^f) \right]^{-1} \sum_{z^f \in D_f} \nabla \ell(h_{\boldsymbol{\theta}_{tr}^*}, z^f) \\
&= \underbrace{\left( \sum_{z^{tr} \in D_{tr}} \nabla^2 \ell(\theta_r^*, z^{tr}) \right)^{-1} \sum_{z^{tr} \in D_{tr}} \nabla \ell(\theta_r^*, z^{tr})}_{\vec{a}} - \underbrace{\left( \sum_{z^r \in D_r} \nabla^2 \ell(\theta_{tr}^*, z^r) \right)^{-1} \sum_{z^f \in D_f} \nabla \ell(\theta_{tr}^*, z^f)}_{\vec{b}}
\end{aligned}
\tag{16}
$$

Thus, $\|\boldsymbol{\theta}_r^* - \boldsymbol{\theta}_f^*\| = 0$ if and only if

$$
\sum_{z^f \in D_f} \nabla_{\boldsymbol{\theta}} \ell(h_{\boldsymbol{\theta}_r^*}, z^f) = - \left( \sum_{z^{tr} \in D_{tr}} \nabla_{\boldsymbol{\theta}}^2 \ell(h_{\boldsymbol{\theta}_r^*}, z^{tr}) \right) \left( \sum_{z^r \in D_r} \nabla_{\boldsymbol{\theta}}^2 \ell(h_{\boldsymbol{\theta}_{tr}^*}, z^r) \right)^{-1} \sum_{z^f \in D_f} \nabla_{\boldsymbol{\theta}} \ell(h_{\boldsymbol{\theta}_{tr}^*}, z^f)
\tag{17}
$$

## C.2 PROOF FOR THEOREM 2

Recall the loss calculation in label smoothing and we have

$$
\ell(h_{\boldsymbol{\theta}}, z^{\text{GLS}, \alpha}) = (1 + \frac{1 - K}{K} \alpha) \ell(h_{\boldsymbol{\theta}}, (x, y)) + \frac{\alpha}{K} \sum_{y' \in \mathcal{Y} \setminus y} \ell(h_{\boldsymbol{\theta}}, (x, y'))),
\tag{18}
$$

where we use notations $\ell(h_{\boldsymbol{\theta}}, (x, y)) := \ell(h_{\boldsymbol{\theta}}, z)$ to specify the loss of an example $z = \{x, y\}$ existing in the dataset and $\ell(h_{\boldsymbol{\theta}}, (x, y'))$ to denote the loss of an example when its label is replaced with $y'$. $\nabla_{\boldsymbol{\theta}} \ell(h_{\boldsymbol{\theta}}, (x, y))$ is the gradient of the target label and $\sum_{y' \in \mathcal{Y} \setminus y} \nabla_{\boldsymbol{\theta}} \ell(h_{\boldsymbol{\theta}}, (x, y'))$ is the sum of the gradient of non-target labels.

With label smoothing in Equation 18, Equation 16 becomes

$$
\begin{aligned}
&\boldsymbol{\theta}_r^* - \boldsymbol{\theta}_{f, \text{LS}}^* \\
&\approx \vec{a} + (1 + \frac{1 - K}{K} \alpha) \cdot (-\vec{b}) \\
&\quad + \frac{1 - K}{K} \alpha \cdot \underbrace{\frac{1}{K - 1} \left( \sum_{z^r \in D_r} \nabla_{\boldsymbol{\theta}}^2 \ell(h_{\boldsymbol{\theta}_{tr}^*}, z^r) \right)^{-1} \sum_{z^f \in D_f} \nabla_{\boldsymbol{\theta}} \sum_{y' \in \mathcal{Y} \setminus y^f} \ell(h_{\boldsymbol{\theta}_{tr}^*}, (x^f, y'))}_{\vec{c}} \\
&= \vec{a} + (1 + \frac{1 - K}{K} \alpha) \cdot (-\vec{b}) + \frac{1 - K}{K} \alpha \cdot \vec{c} \\
&= \vec{a} - \vec{b} + \frac{1 - K}{K} \alpha \cdot (\vec{c} - \vec{b})
\end{aligned}
\tag{19}
$$

where

$$
\vec{a} := \left( \sum_{z^{tr} \in D_{tr}} \nabla_{\boldsymbol{\theta}}^2 \ell(h_{\boldsymbol{\theta}_r^*}, z^{tr}) \right)^{-1} \sum_{z^{tr} \in D_{tr}} \nabla_{\boldsymbol{\theta}} \ell(h_{\boldsymbol{\theta}_r^*}, z^{tr}), \quad \vec{b} := \left( \sum_{z^r \in D_r} \nabla_{\boldsymbol{\theta}}^2 \ell(h_{\boldsymbol{\theta}_{tr}^*}, z^r) \right)^{-1} \sum_{z^f \in D_f} \nabla_{\boldsymbol{\theta}} \ell(h_{\boldsymbol{\theta}_{tr}^*}, z^f)
$$

as given in Equation 16.

So we have

$$
\boldsymbol{\theta}_r^* - \boldsymbol{\theta}_{f, LS}^* \approx \vec{a} - \vec{b} + \frac{1 - K}{K} \alpha \cdot (\vec{c} - \vec{b})
\tag{20}
$$

where $\vec{c} := \frac{1}{K-1} \left( \sum_{z^r \in D_r} \nabla^2_{\boldsymbol{\theta}} \ell(h_{\boldsymbol{\theta}^*_{tr}}, z^r) \right)^{-1} \sum_{z^f \in D_f} \nabla_{\boldsymbol{\theta}} \sum_{y' \in \mathcal{Y} \setminus y^f} \ell(h_{\boldsymbol{\theta}^*_{tr}}, (x^f, y'))$. When

$$\langle \vec{a} - \vec{b}, \vec{c} - \vec{b} \rangle \leq 0, \tag{21}$$

$\alpha < 0$ can help with MU, making

$$\| \boldsymbol{\theta}^*_r - \boldsymbol{\theta}^*_{f,\text{NLS}} \| \leq \| \boldsymbol{\theta}^*_r - \boldsymbol{\theta}^*_f \| \tag{22}$$

# D  EXPERIMENTS

## D.1  ABLATION STUDY

Gradient-mixed and NLS are the main contribution to the MU improvement. We study the influence of gradient-mixed and NLS on UGradSL and UGradSL+ using random forgetting across all classes in CIFAR-10, respectively. Compared with NLS, PLS is a commonly-used method in GLS. We also study the difference between PLS and NLS by replacing NLS with PLS in our methods. The results are shown in Table 6. We can find that gradient-mixed can improve the GA or FT while NLS can improve the methods further.

Table 6: Ablation study of gradient-mixed and NLS using random forgetting across all classes in CIFAR-10. UGradSL can still work without $D_r$, showing the effectiveness of NLS on MU. Gradient-Mixed cannot be removed from UGradSL+ because UGradSL+ without $D_f$ is the same as FT.

|  | Gradient-Mixed | NLS | PLS | UA ($\uparrow$) | MIA ($\uparrow$) | RA ($\uparrow$) | TA ($\uparrow$) | $H$ ($\uparrow$) | RTE ($\downarrow$, min) |
|---|---|---|---|---|---|---|---|---|---|
| GA |  |  |  | 0.56±0.01 | 1.19±0.05 | 99.48±0.02 | 94.55±0.05 | 0.02 | 0.31 |
| UGradSL |  |  | ✓ | **25.20±1.67** | 33.66±2.11 | 76.41±1.59 | 70.15±1.31 | 0.43 | 0.36 |
|  | ✓ |  |  | 0.58±0.00 | 1.18±0.06 | **99.48±0.02** | **94.61±0.05** | 0.02 | 0.46 |
|  | ✓ | ✓ |  | 20.77±0.75 | **35.45±2.85** | 79.83±0.75 | 73.94±0.75 | **0.58** | 0.45 |
|  | ✓ |  | ✓ | 2.02±0.28 | 18.66±0.03 | 98.03±0.37 | 92.15±0.40 | 0.29 | 0.46 |
| Fine-Tune |  |  |  | 1.10±0.19 | 4.06±0.41 | 99.83±0.03 | 93.70±0.10 | 0.05 | 1.58 |
| UGradSL+ | ✓ |  |  | 14.12±0.27 | 18.31±0.07 | **97.31±0.19** | **90.17±10.17** | 0.10 | 3.07 |
|  | ✓ | ✓ |  | **25.13±0.49** | **37.19±2.23** | 90.77±0.20 | 84.78±0.69 | **1.84** | 3.07 |
|  | ✓ |  | ✓ | 10.81±3.76 | 22.29±0.81 | 93.98±3.10 | 87.96±2.68 | 1.20 | 3.01 |

## D.2  BASELINE METHODS

Retrain is to train the model using $D_r$ from scratch. The hyper-parameters are the same as the original training. FT is to fine-tune the original model $\boldsymbol{\theta}_o$ trained from $D_{tr}$ using $D_r$. The differences between FT and retrain are the model initialization $\boldsymbol{\theta}_o$ and much smaller training epochs. FF is to perturb the $\boldsymbol{\theta}_o$ by adding the Gaussian noise, which with a zero mean and a covariance corresponds to the 4th root of the Fisher Information Matrix with respect to (w.r.t.) $\boldsymbol{\theta}_o$ on $D_r$ (Golatkar et al., 2020). IU uses influence function (Koh & Liang, 2017) to estimate the change from $\boldsymbol{\theta}_o$ to $\boldsymbol{\theta}_u$ when one training sample is removed. BU unlearns the data by assigning pseudo label and manipulating the decision boundary.

## D.3  IMPLEMENTATION DETAILS

We run all the experiments using PyTorch 1.12 on NVIDIA A5000 GPUs and AMD EPYC 7513 32-Core Processor. For CIFAR-10, CIFAR-100 and SVHN, the training epochs learning rate are 160 and 0.01, respectively. For ImageNet, the training epochs are 90. For 20 Newsgroup, the training epochs are 60. The batch size is 256 for all the dataset. Retrain follows the same settings of training. For fine-tune (FT), the training epochs and learning rate are 10 and 0.01, respectively. For gradient ascent (GA), the training epochs and learning rate are 10 and 0.0001, respectively.

## D.4  CLASS-WISE FORGETTING

We present the performance of class-wise forgetting in CIFAR-100 and SVHN dataset in Table 7. The observation is similar in CIFAR-10, 20 Newsgroup and ImageNet given in Table 1. UGradSL

Table 7: The experiment results of class-wise forgetting in CIFAR-100 and SVHN.

| CIFAR-100 | UA (↑) | MIA (↑) | RA (↑) | TA (↑) | $H$ (↑) | RTE (↓, min) |
|---|---|---|---|---|---|---|
| Retrain | 100.00±0.00 | 100.00±0.00 | 99.96±0.01 | 74.86±0.19 | 1.85 | 26.95 |
| FT | 19.22±8.02 | 75.36±13.78 | **99.80±0.01** | **75.01±0.37** | 1.84 | 1.74 |
| GA | 65.26±0.68 | 90.81±5.09 | 95.31±1.41 | 70.19±1.78 | 2.50 | **0.06** |
| IU | 72.42±19.63 | 93.42±3.75 | 95.64±1.69 | 71.18±2.40 | 0.15 | 1.24 |
| FF | 19.22±3.37 | 75.36±2.62 | 99.80±3.61 | 75.01±2.52 | 0.41 | 39.90 |
| BU | 62.74±0.51 | 90.44±5.09 | 95.41±1.28 | 70.27±1.70 | 2.41 | 0.55 |
| UGradSL | 66.59±0.90 | 90.96±5.05 | 95.45±1.42 | 70.34±1.78 | 2.55 | 0.07 |
| UGradSL+ | **89.08±3.95** | **98.93±1.45** | 97.88±1.23 | 70.82±1.71 | **2.58** | 3.37 |
| SVHN | UA (↑) | MIA (↑) | RA (↑) | TA (↑) | $H$ (↑) | RTE (↓, min) |
| Retrain | 100.00±0.00 | 100.00±0.00 | 100.00±0.01 | 95.94±0.11 | 0.80 | 37.05 |
| FT | 6.49±1.49 | 99.98±0.04 | **100.00±0.01** | **96.08±0.01** | 1.22 | 2.42 |
| GA | 87.49±1.94 | 99.85±0.09 | 99.52±0.03 | 95.27±0.21 | 1.12 | 0.15 |
| IU | 93.55±2.78 | 100.00±0.00 | 99.54±0.03 | 95.64±0.31 | 0.03 | 0.23 |
| FF | 72.45±44.51 | 77.98±23.99 | 39.36±41.12 | 37.16±39.36 | 0.03 | 5.88 |
| BU | 85.56±3.07 | 99.98±0.02 | 99.55±0.01 | 95.53±0.07 | 1.42 | 3.17 |
| UGradSL | 90.71±4.08 | 99.90±0.16 | 99.54±0.04 | 95.64±0.25 | **1.46** | 0.23 |
| UGradSL+ | **90.71±1.96** | **100.00±0.00** | **100.00±0.01** | 95.93±0.03 | 1.38 | 4.56 |

Table 8: The experiment results of group forgetting across all the classes in CIFAR-10, ImageNet and SVHN.

| CIFAR-10 | UA (↑) | MIA (↑) | RA (↑) | TA (↑) | $H$ (↑) | RTE (↓, min) |
|---|---|---|---|---|---|---|
| Retrain | 8.07±0.47 | 17.41±0.69 | 100.00±0.01 | 91.61±0.24 | 0.08 | 24.66 |
| FT | 1.10±0.19 | 4.06±0.41 | **99.83±0.03** | 93.70±0.10 | 0.04 | 1.58 |
| GA | 0.56±0.01 | 1.19±0.05 | 99.48±0.02 | **94.55±0.05** | 0.02 | 0.31 |
| IU | 17.51±2.19 | 21.39±1.70 | 83.28±2.44 | 78.13±2.85 | 0.01 | 1.18 |
| FF | 2.27±2.83 | 6.58±6.25 | 98.28±0.67 | 91.95±0.71 | 0.18 | 3.15 |
| BU | 0.48±0.07 | 1.16±0.04 | 99.47±0.01 | 94.58±0.03 | 0.02 | 1.41 |
| UGradSL | 20.77±0.75 | 35.45±2.85 | 79.83±0.75 | 73.94±0.75 | **0.58** | 0.45 |
| UGradSL+ | **25.13±0.49** | **37.19±2.23** | 90.77±0.20 | 84.78±0.69 | **1.84** | 3.07 |
| ImageNet | UA (↑) | MIA (↑) | RA (↑) | TA (↑) | $H$ (↑) | RTE (↓, hr) |
| GA | 17.00±0.15 | 32.80±0.10 | **83.07±0.03** | **70.82±0.02** | 0.83 | 0.03 |
| UGradSL | **28.88±0.72** | **39.36±0.65** | 71.17±0.82 | 60.08±0.57 | **1.67** | 0.06 |
| SVHN | UA (↑) | MIA (↑) | RA (↑) | TA (↑) | $H$ (↑) | RTE (↓, min) |
| Retrain | 4.95±0.03 | 15.59±0.93 | 99.99±0.01 | 95.61±0.22 | 0.09 | 35.65 |
| FT | 0.45±0.14 | 2.30±0.04 | **99.99±0.00** | **95.78±0.01** | 0.02 | 2.76 |
| GA | 0.58±0.04 | 1.13±0.02 | 99.56±0.01 | 95.62±0.01 | 0.02 | **0.31** |
| IU | 2.11±1.10 | 6.35±1.42 | 98.21±0.90 | 92.35±1.23 | 0.02 | 1.52 |
| FF | 0.45±0.09 | 1.30±0.12 | 99.55±0.01 | 95.49±0.03 | 0.03 | 6.02 |
| BU | 0.45±0.14 | 1.13±0.05 | 99.57±0.03 | 95.66±0.01 | 0.02 | 4.24 |
| UGradSL | 6.16±0.49 | 26.35±0.40 | 94.24±0.33 | 90.55±0.27 | **0.32** | 0.57 |
| UGradSL+ | **25.05±4.29** | **35.42±2.13** | 92.43±5.93 | 85.36±4.80 | 0.26 | 4.44 |

and UGradSL+ can improve the MU performance with acceptable time increment and performance drop in $D_r$. In addition, UGradSL and UGradSL+ can improve the randomness of the prediction in $D_f$.

## D.5 RANDOM FORGETTING ACROSS ALL CLASSES

We present the performance of class-wise forgetting in CIFAR-10, ImageNet and SVHN dataset in Table 8. The observation is similar in CIFAR-100 and 20 Newsgroup given in Table 2.

Table 9: The experiment results of group forgetting across all the classes in CIFAR-100

| | UA (↑) | MIA (↑) | RA (↑) | TA (↑) | $H$ (↑) | RTE (↓, min) |
|---|---|---|---|---|---|---|
| Retrain | 55.42±1.56 | 78.01±0.19 | 100.00±0.01 | 76.74±0.32 | 0.64 | 22.03 |
| FT | 3.29±1.03 | 15.54±3.35 | **99.96±0.03** | **83.57±0.18** | 0.13 | 1.59 |
| GA | 2.54±0.36 | 8.11±1.19 | 98.43±0.05 | 82.35±0.26 | 0.08 | **0.46** |
| IU | 25.49±10.29 | 29.45±7.74 | 82.75±5.64 | 67.30±4.28 | 0.42 | 1.17 |
| FF | 1.93±0.18 | 7.54±0.96 | 98.42±0.20 | 83.08±0.11 | 0.23 | 2.89 |
| BU | 2.38±0.14 | 5.95±0.09 | 97.43±0.03 | 76.17±0.01 | 0.12 | 1.40 |
| UGradSL | **54.91±3.94** | **68.28±1.84** | 84.53±1.19 | 61.38±1.34 | **1.20** | 0.74 |
| UGradSL+ | 50.65±0.65 | 69.18±1.94 | 99.21±0.40 | 73.60±1.00 | 0.47 | 2.83 |

Table 10: The experiment results of random forgetting within the single class in CIFAR-10

| | UA (↑) | MIA (↑) | RA (↑) | TA (↑) | $H$ (↑) | RTE (↓, min) |
|---|---|---|---|---|---|---|
| Retrain | 45.03±5.38 | 50.11±4.91 | 75.00±0.00 | 67.63±0.80 | 0.40 | 18.14 |
| FT | 1.49±0.19 | 6.92±0.28 | 74.98±0.01 | 70.68±0.09 | 0.07 | 1.15 |
| GA | 79.56±3.85 | 68.88±1.77 | 72.47±1.33 | 62.66±1.59 | 0.81 | **0.10** |
| IU | 0.31±0.03 | 0.44±0.01 | 74.59±0.01 | 70.91±0.00 | 0.01 | 0.53 |
| BU | 78.33±3.33 | **92.90±2.76** | **97.12±0.99** | **84.09±1.28** | 0.82 | 1.36 |
| UGradSL | **88.72±9.79** | 66.16±10.21 | 65.74±5.87 | 56.71±3.61 | **1.44** | 0.17 |
| UGradSL+ | 87.32±4.40 | 67.64±2.98 | 73.54±0.78 | 62.99±1.15 | 0.32 | 2.96 |

## D.6 GROUP FORGETTING ACROSS ALL CLASSES

We present the performance of group forgetting across all classes in Table 9. The observation is similar in the class-wise group forgetting in Table 3.

## D.7 RANDOM FORGETTING WITHIN THE SINGLE CLASS

Random forgetting within the single class is another unlearning paradigm. We present the results of this unlearning paradigm in CIFAR-10 in Table 10. Although BU is better than UGradSL and UGradSL+, BU is not comprehensive among the other forgetting paradigms.

## D.8 MU WITH THE OTHER CLASSIFIER

To validate the generalization of the proposed method, we also try the other classification model. We test VGG-16 and vision transformer (ViT) on the task of random forgetting across all classes and class-wise forgetting using CIFAR-10, respectively. The results are given in Table 11 and 12. The observation is similar in Table 2 and 1, repsectively.

## D.9 THE EFFECT OF THE SMOOTH RATE

We also investigate the relationship between the performance and the smooth rate $\alpha$. We select UGradSL+ using the random forgetting across all classes in CIFAR-10. The results are given in Figure 3. It should be noted that **the hyper-parameter tuning is acceptable using UGradSL and UGradSL+ because the most important metrics are from** $D_f \in D_{tr}$**.** We do not use any extra information from the testing dataset. Our method can improve the unlearning accuracy (UA) without significant drop of testing accuracy (TA).

## D.10 STREISAND EFFECT

From the perspective of security, it is important to make the predicted distributions are almost the same from the forgetting set $D_f$ and the testing set $D_{te}$, which is called Streisand effect. We investigate this effect in the *random forgetting among all the classes* on CIFAR-10 by plotting confusion matrix as shown in Figure 4. It can be found that our method will not lead to the extra hint of $D_f$.

Table 11: The experiment results of random forgetting across all the classes in CIFAR-10 using VGG-16

|  | UA (↑) | MIA (↑) | RA (↑) | TA (↑) | $H$ (↑) | RTE (↓, min) |
|---|---|---|---|---|---|---|
| Retrain | 11.41±0.41 | 11.97±0.50 | 74.65±0.23 | 66.13±0.16 | 0.10 | 9.48 |
| FT | 1.32±0.13 | 3.48±0.13 | **74.24±0.04** | 67.04±0.10 | 0.06 | 0.60 |
| GA | 1.35±0.08 | 2.18±0.66 | 73.95±0.01 | 66.88±0.01 | 0.03 | **0.14** |
| IU | 1.74±0.09 | 2.16±0.61 | 73.96±0.01 | **68.88±0.00** | 0.03 | 0.24 |
| FF | 1.35±0.09 | 2.21±0.58 | 73.95±0.02 | 66.87±0.04 | 0.03 | 1.02 |
| UGradSL | **13.45±0.63** | **11.77±0.54** | 65.05±0.48 | 58.52±0.38 | 0.36 | 0.19 |
| UGradSL+ | 12.41±0.32 | 14.96±0.52 | 65.90±0.52 | 58.58±0.35 | **1.27** | 1.08 |

Table 12: The experiment results of class-wise forgetting in CIFAR-10 using ViT.

| CIFAR-10 | UA (↑) | MIA (↑) | RA (↑) | TA (↑) | $H$ (↑) | RTE (↓, min) |
|---|---|---|---|---|---|---|
| Retrain | 100.00±0.00 | 100.00±0.00 | 61.41±0.81 | 58.94±1.09 | - | 189.08 |
| FT | 3.97±0.87 | 7.60±1.76 | **98.29±0.05** | **80.44±0.22** | 0.04 | 2.99 |
| GA | 33.77±6.36 | 40.47±6.63 | 89.47±4.21 | 71.65±2.79 | 0.16 | 0.32 |
| UGradSL | 68.11±11.03 | 73.84±9.58 | 84.11±2.70 | 68.33±1.69 | **0.24** | **0.22** |
| UGradSL+ | **99.99±0.01** | **99.99±0.02** | 94.46±1.06 | 77.26±1.19 | 0.01 | 5.86 |

## D.11 GRADIENT ANALYSIS

As mentioned in Section 4.2, $\langle \vec{a} - \vec{b}, \vec{c} - \vec{b} \rangle \leq 0$ always holds practically. We practically check the results on CelebA dataset. The distribution of $\langle \vec{a} - \vec{b}, \vec{c} - \vec{b} \rangle$ is shown in Figure 5, which is with our assumption.

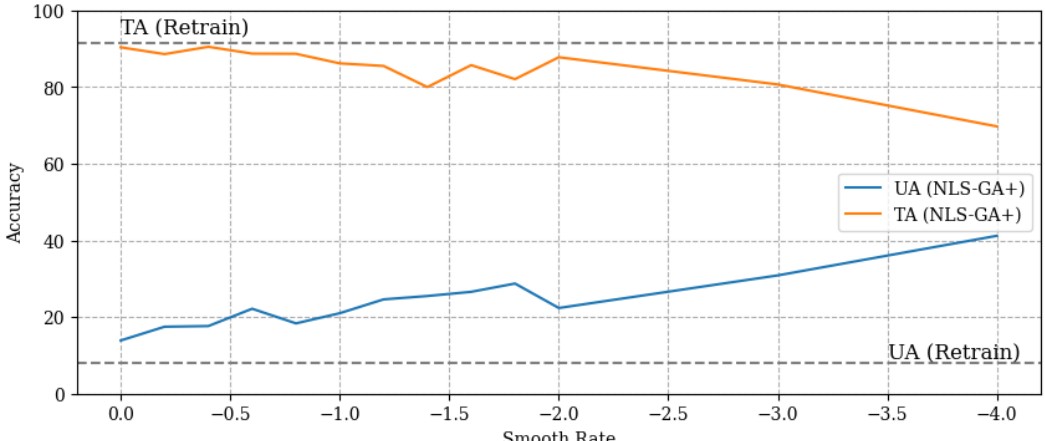

Figure 3: The relationship between the performance and smooth rate in random forgetting across all classes using CIFAR-10. The gray dash line stands for the performance of retrain. Our methods can improve the unlearning accuracy (UA) without significant drop of testing accuracy (TA).

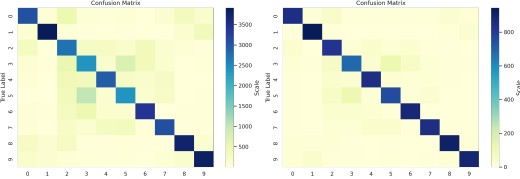

Figure 4: The confusion matrix of testing set and forgetting set $D_f$ using our method on CIFAR-10 with random forgetting across all the classes. There is no big difference between the prediction distribution. Our method will not make $D_f$ more distinguishable.

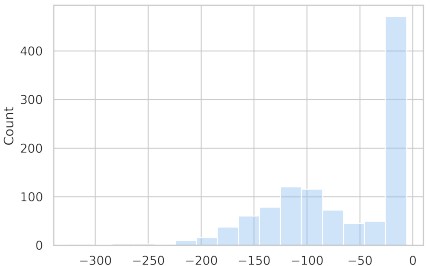

Figure 5: The distribution of $\langle \vec{a} - \vec{b}, \vec{c} - \vec{b} \rangle$ on CelebA dataset.

