# OpenReview forum: "UGradSL: Machine Unlearning Using Gradient-based Smoothed Label"
_ICLR.cc/2024/Conference — Submitted to ICLR 2024_

### Official Review · Reviewer_4Lns · 2023-10-29

**Soundness:** 1 poor
**Presentation:** 2 fair
**Contribution:** 2 fair
**Rating:** 3
**Confidence:** 4

**Summary:**

The authors propose to leverage label smoothing to enhance machine unlearning.

**Strengths:**

-	The label smoothing for unlearning seems interesting.

**Weaknesses:**

-	The introduction on prior machine unlearning problem is incorrect.
-	The evaluation of machine unlearning efficacy is incorrect.

My main concern with this paper is that the authors seem to not understand the machine unlearning problem correctly. First, note that the exact unlearning means that the unlearned model parameters are identical to the one obtained from retraining from scratch in distribution. This is correctly stated by the authors in the Definition 1. However, this is **not** what works that incorporate DP principle achieve in general such as Guo et al. 2019 and Sekhari et al. 2021. By their definition of unlearning, they only ensure that the distribution of their unlearned model weights are **approximately** the same as the retraining one. The approximation error is controllable and there is an inherent utility-privacy-complexity trade-off. It is incorrect to state that these methods are exact unlearning methods and claim that these methods cannot balance between unlearning efficacy (privacy) and computational complexity.

I am also confused with the claim made right after Definition 1. The authors mention that `` The MU performance of retrain using iterative training is sensitive to different hyper-parameters and the distribution of the prediction on $D_f$ is not random enough as given in Section 5, showing that retrain is effectively an approximation of exact MU.`` How does it even make sense when in definition 1 the authors literally define retraining as the gold standard of exact unlearning? I believe the statement of the authors here is not what they really mean, but it is very confusing to readers.

Another issue I have is that the evaluation metric on unlearning efficacy (privacy) is **problematic**. As defined by the authors themselves (definition 1), the optimal case in machine unlearning (at least from the privacy aspect) is retraining. In the experiment section, the authors test several metrics Membership Inference Attack (MIA) and Unlearn Accuracy (UA) as metrics for privacy. The authors claim that higher MIA and UA imply better privacy (unlearning efficacy) for the unlearning method. This is absolutely **incorrect**. Intuitively, consider the following hypothesis test: the given model is from retraining (null hypothesis) or not. Note that if one can achieve very low type I and II errors in this hypothesis test, they can equivalently know that the data ``was’’ seen by the model and thus breach the privacy. As a result, one should measure the unlearning efficacy by how close these metrics are with respect to the retraining model, instead of making them as high as possible. This is exactly the evaluation of Golatkar et al. 2020 and many more unlearning for deep neural networks literature. I also encourage the authors to check the manuscript of the NeurIPS 2023 machine unlearning competition written by Google, where comparing with retraining is their standard as well. As a result, I do not think the privacy-related conclusion drawn from Tables 1 and 2 is correct.

For the same reason, having higher entropy $H$ on the prediction probability of $D_f$ is also not a reasonable metric for privacy evaluation. Note that it is possible that even if we completely remove a sample, the model with perfect privacy having $H\rightarrow0$ still makes sense. Consider the following simple but extreme scenario, where our dataset contains data points that are **identical** to the centroid of each class and they are well-separated (well-clustered). In this case, even if you remove one sample and retrain from scratch, the decision boundary will not change so the retrained model will still be confident (low $H$) on the sample subjected to be removed. In this scenario, having $H$ being unreasonably high is in fact **hurting** the privacy as it will tell the adversary that we do something other than retraining. I hope this simple counter-example can let the authors understand my point.

**Questions:**

Please let me know if I missed something regarding my statement in the weaknesses section.

---

> ### Author Response · Authors · 2023-11-15
> **Response from the author**
>
> 1. Thanks for your clarification of these two retraining definitions. First, as we state in Section 3, not all of the retrained models are exact MU because the convergence of the deep learning model is hard to determine. **Given a fixed forgetting set and model, the prediction distribution on $D_f$ from the exact MU model should be determined**. Here, we want to state that retrained-based MU is hard to balance between unlearning efficacy and computational complexity. In our next version, we will revise these sentences to make them more precise.
>
> 2. As mentioned above, one (not all) of the retrain models can be the exact MU model, which is what Definition 1 tells us. For example, we can train CIFAR-10 using ResNet for 1, 10, and 100 epochs. They are all from the distribution of the retraining model. We don’t think they are all exact MU, or maybe none of them are exact MU. As stated above, given a fixed forgetting set and model, the prediction distribution on $D_f$ from the exact MU model should be determined. The exact MU should treat the forgetting dataset as never seen before at the sample level. Please note that we have a sentence below Definition 1 showing that retrain is effectively an approximation of exact MU, meaning that most retrain is effectively an approximation of exact MU, which is not exact MU because exact MU is very hard to reach and determine in the deep learning scenario. Exact MU is a further case of the retrain model in general.
>
> 3. Thanks for pointing it out. First, UA is not for privacy efficacy. The MIA we use is from [1]. A higher MIA means less information on $D_f$ remains in the unlearned model. Then, we want to state that not all the retrained models can be regarded as exact MU, as we mentioned above. The real exact MU is hard to determine. In addition, how we can evaluate this in real cases should be considered if we don’t have retrained models like ChatGPT, DALLE-2, etc. If we always rely on the retrained model to evaluate the MU methods or select the hyper-parameters, the practical meaning of MU will be much less meaningful. If we cannot access a retrained model in real life, it is hard to compare with the retrained model even to claim privacy leakage. For the cases that cannot access a retrained model, [2, 3] also gives up the retrained model baseline to compare, especially in diffusion models.
>
> We also consider the privacy issue for this “the higher, the better” metric. We compare the distribution difference between the forgetting and testing sets, which should be similar because both datasets are unseen datasets for the unlearned model. Our method shows a similar distribution in Section D.10 in the Appendix.
>
> [1] Jia, Jinghan, et al. "Model Sparsity Can Simplify Machine Unlearning." Thirty-seventh Conference on Neural Information Processing Systems. 2023.
>
> [2] SalUn: Empowering Machine Unlearning via Gradient-based Weight Saliency in Both Image Classification and Generation https://openreview.net/forum?id=gn0mIhQGNM&noteId=E7HA5uwCaw
>
> [3] Forget-Me-Not: Learning to Forget in Text-to-Image Diffusion Models https://openreview.net/forum?id=PuRhqpBTmj
>
>
> We have read the competition instructions, and the organizer uses 512 retrained models to evaluate the $\mathcal{F}$. It is perfect for the ideal cases. As stated in Google's doc, their evaluation metrics may not be a perfect way. We also admit that our evaluation metrics may need improvement. However, we hope our insight and design of the evaluation metrics can help.
>
> 4. Thanks for pointing it out. I understand your perspective is from privacy protection, making forgetting datasets more distinguishable. The entropy metrics are from the information theory perspective. The extreme of forgetting is to randomly guess, which is our idea about exact machine unlearning. For exact MU, the model should behave as not seeing the sample completely, which should randomly guess. As for your concerns about privacy protection, we also have the experiment shown in Section D.10 in the Appendix. The prediction on the forgetting dataset and testing dataset are almost the same. They are both unseen datasets, which should be reasonable.

---

> > ### Comment · Reviewer_4Lns · 2023-12-02
> >
> > I thank the rebuttal from the authors. However, I am still not convinced by the explanation.
> >
> > 1. Note that retrained models are indeed exact unlearning no matter their convergence. For instance, even if the deep learning model has two equally well optimums and retraining will converge to either one of them with a probability of 0.5. The underlying model parameter distribution is still well-defined. This is also why the organizer of the NeurIPS 2023 machine unlearning competition has to do multiple independent trials. If the authors do not want to compare with retraining, they should specifically say that their goal is not for privacy, otherwise, the current argument is still problematic.
> >
> > 2. The authors should use "exactly the same hyperparameter" for retraining, otherwise the choice of hyperparameter will obviously breach the privacy. This is the definition of retraining, see the previous theoretical machine unlearning literature.
> >
> > 3. This is again not justifying the authors' approach. While I agree that retraining can be hard to obtain for large models like ChatGPT, it does not mean we can rely on an unjustified criterion. Evaluation unlearning without retraining is indeed an important problem, and I believe it is still open to the best of my knowledge. The authors should be more careful in justifying the proposed criterion, especially in privacy problems. I would encourage the authors to simply work on unlearning that is not for privacy if retraining is not an option.
> >
> > 4. As I mentioned, I encourage the authors to define their unlearning problem not in terms of privacy. Note that the sentence " The extreme of forgetting is to randomly guess" is not correct in terms of privacy, as in my example.

---

### Official Review · Reviewer_32Fz · 2023-11-01

**Soundness:** 3 good
**Presentation:** 3 good
**Contribution:** 2 fair
**Rating:** 3
**Confidence:** 5

**Summary:**

The authors propose an unlearning method by using negative label smoothing as the unlearning algorithm. They show that the proposed method can reduce the confidence for the forgotten samples thereby attempting to achieve unlearning. The paper uses theoretical inspiration from linearization of the network along with influence functions to provide theoretical arguments. Finally the paper provide empirical evidence to support their claims.

**Strengths:**

1. The paper address the problem of unlearning which a pretty important problem given the increase in use of foundation models.
2. The proposed algorithm is easy to implement, and does not require computation of any hessians.
3. For a linear model, the proposed theorems are very intuitive, and provide insights about the unlearning procedure.
4. The authors have performed extensive empirical evaluation

**Weaknesses:**

1. The theoretical analysis in the paper is based on influence functions, however it is well know that influence functions dont work well for deep neural networks [1]. This makes the theoretical foundation of the work weak.
2. Forgetting based on linearization of deep networks was already studied in [2,3] which is not cited in the paper, nor is the method compared against.
3. [3] proposed a method where they linearized a network, such that the optimization problem is convex, such an approach could be extremely useful with the theoretical framework provided in the paper, as it would make the theorems more applicable and rigorous.
4. The proposed method is an approximate unlearning method, however the paper does not provide a bound on the error as a function of unlearning requests. This is an imperative result for approximate unlearning algorithms.
5. To make the theoretical results be more compatible with the empirical results the authors can consider fine-tuning (as the model will be more convex) compared to training from scratch where the landscape is highly non-convex.
6. It will be interesting to see how the weights in different layers move after learning, i suspect that the last layer moves the most while the lower layers wont change much in value, this could correspond to unlearning in the activations but not the weights. Having such a plot could be interesting for analysis.
7. [4] has shown that the algorithm and definition proposed by Bourtoule et al may not be sufficient against adaptive unlearning requests, which may require shades of DP( differential privacy) in the unlearning algorithm How do you plan to add this to the existing approach?
8. All the empirical results measure the unlearning in the final activations of the model, which does not consider unlearning from the weights of the model. How do you ensure that the information is removed from the weights? and how do you plan to show it empirically?

[1]Influence Functions in Deep Learning Are Fragile, https://arxiv.org/abs/2006.14651
[2]Forgetting Outside the Box: Scrubbing Deep Networks of Information Accessible from Input-Output Observations(https://arxiv.org/abs/2003.02960)
[3]Mixed-Privacy Forgetting in Deep Networks(https://arxiv.org/abs/2012.13431)
[4]Adaptive Machine Unlearning(https://arxiv.org/abs/2106.04378)

**Questions:**

See weaknesses above.

---

> ### Author Response · Authors · 2023-11-22
> **Response from the author**
>
> 1. Thanks for your suggestion. It should be noted that our framework only considers the converged weight. What we did is to make the Taylor expansion on these converged weights as shown in Equations 6 and 7. The converged model weight can be seen as locally convex. We did not consider any model weight which is not converged.
>
> 2. Thanks for your advice. We will add these references and compare these methods in our next version.
>
> 3. Thanks for your advice. We will look at them very carefully and seriously consider integrating the framework into ours in our next version.
>
> 4. Thanks for your question. Our method is based on the existence of ideal retraining in close form. This expression is hard to get. The current widely accepted definition is given in Definition 1. It is hard for us to provide the error range based on the expression in Definition 1.
>
> 5. Thanks for your advice. UGradSL+ is a method based on the finetune. Actually, all our methods are based on the pre-trained model. Thus, it can be regarded as locally convex as what we mention in the first answer.
>
> 6. Thanks for your advice. We will seriously consider adding this plot in our next version.
>
> 7. Thanks for your advice. Our settings is not based on the adaptive unlearning framework. But your suggestion can be a direction of our future work.
>
> 8. First, evaluating the unlearning performance based on weight is unreasonable. According to [1], the same output can be from two models with totally different weights. Thus, a model which is not unlearned is also likely to predict similar results as the one unlearned. That's the reason why we use the output rather than the weight. As for the information removal, as we mentioned in Evaluation Metrics and [2], A higher MIA-Efficacy implies less information about $D_f$ in $\theta_u$ where $D_f$ is the forgetting set and $\theta_u$ is the unlearned model weight.
>
> [1] Thudi, Anvith, et al. "On the necessity of auditable algorithmic definitions for machine unlearning." 31st USENIX Security Symposium (USENIX Security 22). 2022.
>
> [2] Jia et al. Model sparsification can simplify machine unlearning[J]. arXiv preprint arXiv, 2023.

---

### Official Review · Reviewer_RaEn · 2023-11-01

**Soundness:** 2 fair
**Presentation:** 3 good
**Contribution:** 2 fair
**Rating:** 3
**Confidence:** 2

**Summary:**

This paper focuses on the machine unlearning problem and proposes a gradient-based unlearning algorithm. Targeting the high time cost and significant drops of remaining and test accuracies, this paper proposes to use the label smoothing technique in the design of the unlearning loss function. This paper firstly proves that the gradient ascent can achieve exact unlearning and then it proves that using the generalized label smoothing in the gradient ascent can narrow down the errors. Then this paper proposes UGradSL and UGradSL+ algorithms for unlearning, which iterate over the remaining and forgetting dataset respectively. Then this paper sets up experiments on class unlearning, randomly selected data unlearning, and sub-class unlearning. The experiments present improvements compared with the selected baseline methods.

**Strengths:**

a) This paper proposes a different direction of reducing the performance drop of remaining and test data by label smoothing.

b) This paper provides both theoretical and experimental evidence to support the proposed method.

c) The experiments are comprehensive, including three different unlearning tasks and multiple datasets ranging from image to text data. The time efficiency of the proposed method is significantly improved.

**Weaknesses:**

a) This paper assumes that “the exact MU should behave as the model does not access $D_f$ at all so that the prediction on $D_f$ should be as random as  possible.” This paper cannot hold for most of the unlearning scenarios. For example, in the sub-class unlearning as this paper mentions, if the forgetting sub-class is fighter aircraft and the super-class is the airplane, the data of fighter aircraft should also be more likely to be classified as the airplane after unlearning because the fighter aircraft and other remaining planes share many similar features. In addition, in your definition of unlearning, $\Theta_r$ is the model distributions learned from the remaining data and it does not have any restrictions on the forgetting data.

b) The proof of exact unlearning in theorem 1 looks weird: this paper uses the Taylor series and uses the approximately equal symbol in eq.(6) and eq.(7). Thus, how to get a conclusion with the strict equal symbol of the exact unlearning in theorem 1.

c) Some issues exist in the presentations, like notations and terminologies. For example, the term Unlearning Diagram is not explained either on page 7 or the appendix reference is missing on page 9.

d) This paper claims that the group unlearning is one of the contributions of this paper. However, this type of unlearning task has been discussed in [1]

e) The evaluation part of this paper mentions that the higher unlearning errors, remaining accuracies, and test accuracies stand for the better unlearning performances for all unlearning tasks. However, some other papers like [2] which this paper cites, [3], and [4], all mention that the closer to the models that are trained on the remaining data will be better.

[1] Chundawat et al. Can bad teaching induce forgetting? unlearning in deep networks using an incompetent teacher. AAAI 2023.

[2] Jia et al. Model sparsification can simplify machine unlearning[J]. arXiv preprint arXiv, 2023.

[3] Zhang et al. Prompt certified machine unlearning with randomized gradient smoothing and quantization. NeurIPS, 2022.

[4] Golatkar et al. Eternal sunshine of the spotless net: Selective forgetting in deep networks. CVPR. 2020

**Questions:**

a) Why use the approximately equal symbol in eq.(8)?

b) How to understand if the model outputs random predictions, ⟨a-b,c-b⟩ ≤ 0 will
hold?

c) What are the training details of the proposed methods, especially for the alpha, E, and p? Does this paper use any pre-train parameters for the models?

d) This paper has mentioned 5 baselines. What are the standards for selecting
baselines and why some results of these baselines are missing in experiments?

---

> ### Author Response · Authors · 2023-11-22
> **Response from the author**
>
> Weakness:
>
> (a) Thanks for your question. First, we argue that not all the retrain models can be seen as exact MU models because retraining based on deep learning in an iterative style is very sensitive to hyperparameters. It is hard to determine whether the mode is converged or not. Given the forgetting dataset D_f and the model, the exact MU model should be determined, which should not be influenced by the hyper-parameters. From the theory of information, the unknown status of one data point should be random, whose entropy is highest.
>
> We admit that your example is valid because of the model generalization. However, the current unlearn is still an approximate MU rather than the real exact MU. Again, from the theory of information, the exact MU predicts the unknown data point should be random.
>
> (b) Thanks for your advice. We will revise it in our next version.
>
> (c) Thanks for your advice. We will recheck the notation and terminologies in our next version. As for the Unlearning Diagram, it is a typo of the Unlearning Paradigm.
>
> (d) Thanks for your advice. We will modify the contribution list.
>
> (e)  Thanks for your question. We have noticed that references [2-4] do use a retrained model as the baseline. However, it should be noted that we often cannot access the retrained model in real usage, especially for industry-level large language models. Some of the methods are sensitive to hyperparameters like the number of epochs, learning rate, etc. It is impossible to find a more suitable parameter that makes the performance of the unlearned model closer to the retrained model because we don’t have the retrained model in real usage. Thus, we believe that we cannot use the performance of the retrained model as a baseline. For the reason why the higher is the better, it is natural to have the unlearned model hurt the testing or remaining performance as little as possible. For the classification task, if the fewer data points the unlearned model can remember, the higher the unlearning errors and the better the unlearning is.
>
> Questions:
>
> (a) Because Eq. 8 is the subtraction of Eq. 6 and 7. Eq. 6 and 7 are the approximations of Taylor Series which are approximately equal. We will revise this statement in our next version.
>
> (b) Thanks for your question. First, a, b, and c are vectors here with directions. Random outputs mean that the gradient of the target label and the non-target label are diverse or not in the same direction because every time, the largest gradient will point to different labels (directions), which is what <a-b, c-b> tells us.
>
> (c) For different datasets, the alpha, E and p are different which shows the different difficulties to unlearn. For example, class-wise unlearning should be easier than random unlearning. Alpha should be smaller in random unlearning. But it should be noted that our method does not rely on the unseen data because our metrics are higher, the better. Thus, it is suitable for the method always to find a better parameter as given in Section D.9 in the Appendix.
>
> No. Our method only relies on the pre-trained model and does not rely on the pre-trained parameters.
>
> (d) We selected the baseline methods mainly based on [1] and boundary unlearn, which is accepted in CVPR 2023. The reason why we have some missing values is that some of the methods only work on one modality or the method is too slow for a very large dataset like ImageNet, CelebA. Our methods are general for both vision and language.
>
> [1] Jia et al. Model sparsification can simplify machine unlearning[J]. arXiv preprint arXiv, 2023.

---

### Meta-Review · Area_Chair_nw54 · 2023-12-05

**Metareview:**

This paper presents a method for machine unlearning (MU) using smoothed labels. The paper proposed a plug-and-play method along with theoretical analysis of the same.

The method is simple and easy to implement. However, the reviewers raised several concerns, such as the theoretical foundation based on influence function being weak, the linearization based forgetting idea to be not novel (missing reference as well), the evaluation to be problematic, and lacking bound on the error which is essential for approximate unlearning methods like the one the paper proposes.

The authors submitted a rebuttal but the concerns lingered.

Based on the reviews, rebuttal, the follow-up response from some of the reviewers, and my own reading of the paper, the paper seems to have several serious issues which need to be addressed. The authors should consider the feedback from the reviewers, address the issues raised, and consider resubmitting to another venue.

**Justification For Why Not Higher Score:**

The paper has several technical issues as pointed out by the reviewers and also summarized in the meta-review.

**Justification For Why Not Lower Score:**

N/A

---

### Decision · Program_Chairs · 2024-01-16

Reject